# Adaptive Estimation and Learning under Temporal Distribution Shift

**Dheeraj Baby** [1]  **Yifei Tang** [1]  **Hieu Duy Nguyen** [1]  **Yu-Xiang Wang** [1][2]  **Rohit Pyati** [1]

## Abstract

In this paper, we study the problem of estimation and learning under temporal distribution shift. Consider an observation sequence of length $n$, which is a noisy realization of a time-varying groundtruth sequence. Our focus is to develop methods to estimate the groundtruth at the final time-step while providing sharp point-wise estimation error rates. We show that, *without prior knowledge* on the level of temporal shift, a wavelet soft-thresholding estimator provides an *optimal* estimation error bound for the groundtruth. Our proposed estimation method generalizes existing researches Mazzetto and Upfal (2023) by establishing a connection between the sequence's non-stationarity level and the sparsity in the wavelet-transformed domain. Our theoretical findings are validated by numerical experiments. Additionally, we applied the estimator to derive sparsity-aware excess risk bounds for binary classification under distribution shift and to develop computationally efficient training objectives. As a final contribution, we draw parallels between our results and the classical signal processing problem of total-variation denoising (Mammen and van de Geer, 1997; Tibshirani, 2014a), uncovering *novel optimal* algorithms for such task.

## 1. Introduction

Standard statistical estimation problems assumes access to independent and identically distributed (i.i.d.) observations. However, in many practical applications the familiar i.i.d. assumption is often not applicable. Hence it is important to study the statistical estimation problem while relaxing such assumptions. In this paper, we take a step in this direction by relaxing the identical distributed assumption on the observations.

Concretely, we consider an estimation task where we are given access to $n$ independently drawn observations $y_n, y_{n-1}, \ldots, y_1$ such that $E[y_i] = \theta_i \in \mathbb{R}$ and $\mathrm{Var}(y_i)$ being finite for all $i \in [n] := \{1, \ldots, n\}$. Our goal is to construct an estimate $\hat{\theta}_1$ for the most recent ground-truth $\theta_1$. This estimation task is strongly related to various practical scenarios, such as estimating/predicting the latest stock price, temperature, humidity or network traffic based on a sequence of historically observed and correlated measurements. We remark that any algorithm for the aforementioned problem setting can be directly used to construct estimates for $\theta_n, \ldots, \theta_2$ as well. The reverse-indexing of time is mainly used for notational convenience.

Note that constructing an estimator $\hat{\theta}_1$ itself is not enough. In many scenarios, we also need to provide statistical *guarantees* for such estimators, for example to calculate the risk profile of a stock-picking strategy for a stock-price estimator. To achieve very high estimation quality, we would like to obtain a point-wise performance guarantee for the estimate, i.e., a bound on $|\hat{\theta}_1 - \theta_1|$ which is as small as possible. This requirement is more rigorous in comparison to controlling the estimation error for the sequqnce $\theta_n, \ldots, \theta_1$ in terms of some aggregate performance metric. For example, a point-wise bound implies a control over cumulative performance metrics like the mean squared error (MSE) or regret from online learning (Hazan, 2016) is controlled.

A natural algorithm for estimating $\theta_1$ is to average the most recent observations within a sliding window. However deciding on the optimal window size in a completely data-adaptive manner is non-trivial. Choosing a small window size will lead to an estimate with small bias but large variance. A large window size will lead to variance reduction at the expense of introducing large bias. Hence to attain a sharp point-wise estimation error guarantee, one must choose a window size with an optimal bias-variance trade-off. The second challenge for this task is the dependency on the smoothness/stationarity of the ground-truth sequence near the timepoint 1 of interest, which directly influences the optimal window. In practice, we often do not have prior knowledge on the smoothness/stationarity of the ground-truth sequence. As a consequence, we also need to (potentially in an implicit way) learn the smoothness/stationarity of the underlying ground-truth sequence and adjust the averaging window size accordingly.

---

[1]Amazon [2]University of California San Diego. Correspondence to: Dheeraj Baby <dheerajbaby@gmail.com>.

*Proceedings of the 42$^{nd}$ International Conference on Machine Learning*, Vancouver, Canada. PMLR 267, 2025. Copyright 2025 by the author(s).

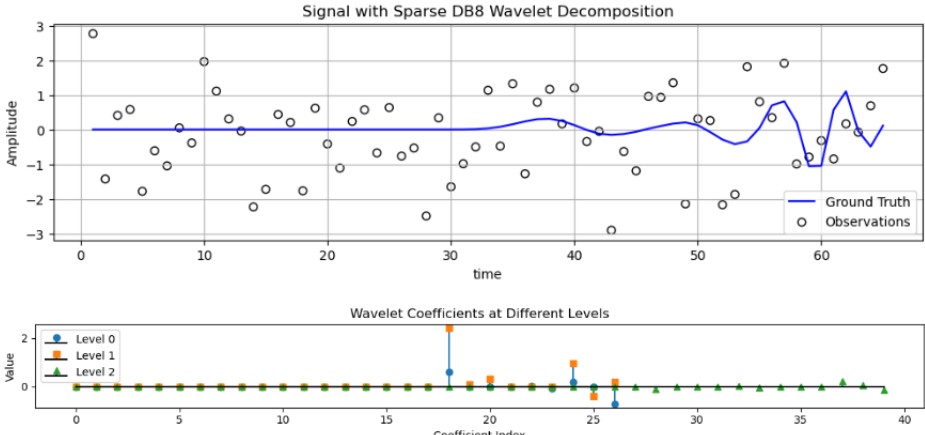

Figure 1: *We have access to noisy observations of a ground truth signal, and our task is to estimate the ground truth at the latest time. Although the ground truth signal appears to exhibit a non-stationary trend in the time domain, the wavelet transform reveals a sparse set of coefficients. As a result, we only need to estimate a few key coefficients, enabling wavelet denoising-based estimators to achieve sharp point-wise and data-adaptive error rates.*

A version of this problem has been posed as an open problem in Hanneke and Yang (2019) and was first addressed in the work of Mazzetto and Upfal (2023). In this study, we take a fresh look at the same problem and show that the famous wavelet soft thresholding algorithm (Donoho, 1995), using Haar wavelets, already recovers the results from Mazzetto and Upfal (2023). The solution put forward by this paper based on the general machinery of wavelets not only offers fresh perspectives but also lead to deeper statistical implications on the estimation problem, which are described as follows.

Specifically, as remarked earlier, the quality of estimation is connected to the stationarity (or smoothness) level of the ground-truth sequence. The more stationary the sequence is, sharper the guarantees/bounds of the estimate become. The solution in Mazzetto and Upfal (2023) leverages local stability to capture non-stationarity, i.e, their algorithm effectively chooses the largest window in which the ground-truth sequence (recall that the sequence formed by starting from time 1 and going to less recent indices) has small variation and is close to a constant signal. The larger such a window is, the smaller the estimation error rate becomes since averaging with such a window can lead to significant variance reduction while introducing only a little bias. Approaching the problem from a different angle, we show that an estimator, based on wavelet soft thresholding, will result into an estimation error which strongly correlates with the sparsity of the wavelet coefficients of the ground-truth sequence. From that perspective, the stationarity level of the ground-truth is strongly connected to the sparsity level of the wavelet coefficients. Our approach based on wavelet coefficients' sparsity is more general than the notion of local stability. For example, by using higher order Daubechies wavelets, one can potentially capture complex trends in

the evolution of the groundtruth sequence with a sparse set of wavelet coefficients, leading to fast estimation guarantees/bounds (see Fig. 1). On the other hand, in the presence of complex temporal evolution patterns, the optimal window chosen based on the notion of local stability can potentially be small which cause a moving-average estimate to incur larger error due to high variance.

We summarize our list of contributions below:

- We show that a wavelet-denoising-based algorithm using Haar tranform achieves the optimal *point-wise* estimation error guarantees similar to that of Mazzetto and Upfal (2023) (Theorem 2). Note that by using higher order wavelet transforms, it is possible to obtain a general error bound, based on the sparsity level of the wavelet coefficients of the ground-truth, which can be potentially sharper than the bounds obtained by using the Haar system (Lemma 1).

- We present a rigorous theoretical examination of distribution shift's consequences for machine learning models, conclusively showing that temporal variations undermine model performance. (Section 3).

- We consider the problem of binary classification under temporally shifted dataset. By using wavelet-based loss estimators, we obtain an oracle-efficient and statistically near-optimal algorithm that requires a *single* call to empirical risk minimization (ERM) oracle, thereby improving the computational efficiency in comparison to the algorithm proposed in Mazzetto and Upfal (2023) which requires $O(\log n)$ ERM calls (Theorem 9). Furthermore for differentiable surrogate losses, our ERM objective preserves the differentiability and en-

ables oracle-efficient procedures, unlike the methods in prior works (Section 4).

- We prove a general result that any algorithm obtaining a point-wise estimation error guarantee similar to that from our proposed wavelet denoising algorithm under Haar system (see Theorem 2) is automatically minimax optimal for the problem of total-variation (TV) denoising. This reveals a previously unknown minimax optimality of several existing algorithms for the TV denoising problem (Section 5).

- Finally, we present numerical results to showcase the superior performance of our proposed (higher order) wavelet-denoising based algorithms in estimating the ground-truth, compared to prior works (Section 6.1).

The rest of the paper is organized as follows: Section 2 presents our theory and algorithm for estimating the ground truth. Section 3 analyzes train-test distribution mismatch, followed by a binary classification algorithm for temporally drifted data in Section 4. Connections to TV-denoising are explored in Section 5, and experimental results are in Section 6.1. Due to space constraints, discussion on related works and preliminaries are deferred to Appendix A and B respectively.

## 2. Estimation Under Temporal Distribution Shift

In this section we consider the estimation setup specified in Section 1 and study algorithmic approaches to estimate the groundtruth at the latest time. This will later help to train ML models under training distribution shift in Section 4.

### 2.1. Algorithm

We propose to use a wavelet-denoising-based algorithm, which is adapted from the well-known idea of wavelet smoothing (Donoho, 1995). The algorithm is presented in Algorithm 1 for the sake of completeness. A fundamental difference between the wavelet-based solution and algorithms from prior works (Mazzetto and Upfal, 2023; Han et al., 2024) is that while the latter maintain an adaptive window-size for averaging the relevant past observations, the wavelet-based solution maintains no such window-size. Instead it implicitly uses the most relevant portion of data to estimate the groundtruth at the final time-step.

### 2.2. Analysis

All proofs of this section are deferred to the Appendices. We emphasize that all the estimation error bounds presented in this section are achievable by Algorithm 1 without any prior knowledge on the degree of distribution drift. We begin

---

**Algorithm 1** Wavelet-Denoising Algorithm

1: **Input:** data $y_n, \ldots, y_1 \in \mathbb{R}^d$, Wavelet Transform matrix $\boldsymbol{W}$, soft-threshold $\lambda$, failure probability $\delta$.
2: Initialize $\boldsymbol{y} \leftarrow [y_n, y_1, \ldots, y_1]^T \in \mathbb{R}^n$.
3: Compute empirical wavelet coefficients $\tilde{\boldsymbol{\beta}} \leftarrow \boldsymbol{W}\boldsymbol{y}$.
4: Compute denoised coefficients $\hat{\boldsymbol{\beta}} \leftarrow T_\lambda(\tilde{\boldsymbol{\beta}})$, where for an $x \in \mathbb{R}, T_\lambda(x) := \text{sign}(x)\max\{|x| - \lambda, 0\}$ is the soft-thresholding operator. When acted upon a vector, the soft-thresholding is performed coordinate-wise.
5: Reconstruct (a.k.a inverse wavelet transform) the signal by $\hat{\boldsymbol{\theta}} \leftarrow \boldsymbol{W}^T\hat{\boldsymbol{\beta}}$.
6: Return the last coordinate of $\hat{\boldsymbol{\theta}}$.

---

with a simple property of soft-thresholding operation.

**Lemma 1.** *Consider the observation model $y_i = \theta_i + \epsilon_i$, for $i = 1, \ldots, n$ with $\epsilon_i$ being iid $\sigma$-sub-gaussian random variables. Let $\boldsymbol{y} := [y_n, \ldots, y_1]^T$, $\boldsymbol{\theta} = [\theta_n, \ldots, \theta_1]$ and $\boldsymbol{W}$ be an orthonormal wavelet transform matrix. Let $\tilde{\boldsymbol{\beta}} := \boldsymbol{W}\boldsymbol{y}$ and $\boldsymbol{\beta} := \boldsymbol{W}\boldsymbol{\theta}$ be respectively the empirical and true wavelet coefficients. Let $\mathcal{I}$ be an index set of wavelet coefficients that affect the value of reconstruction of the last groundtruth $\theta_1$. Let $W_{i,n}$ be the value of the element in $i^{th}$ row and $n^{th}$ column of $\boldsymbol{W}$. Let $\hat{\theta}_1$ be the estimate of the groundtruth $\theta_1$ obtained via Algorithm 1 with $\lambda = 2\sigma\sqrt{2\log(\log n/\delta)}$. Define $(a \wedge b) := \min\{a, b\}$, we have with probability at-least $1 - \delta$ that*

$$|\hat{\theta}_1 - \theta_1| \leq \sum_{i \in \mathcal{I}} 6|W_{i,n}| \cdot (|\beta_i| \wedge \lambda).$$

Next we show that if we use the Haar wavelet system in Algorithm 1, one can recover the variational bounds similar to the results in Mazzetto and Upfal (2023), indicating the versatility of the wavelet-based solution.

**Theorem 2.** *Let $\bar{\theta}_{t:1} = (\theta_1 + \ldots + \theta_t)/t$. For a time-point $r$, let $S(r) := \{1, 2, 4, \ldots, 2^{\lfloor \log_2 r \rfloor}\}$. Let $U(r) := \max_{t \in S(r)} |\bar{\theta}_{t:1} - \theta_1|) \vee \sigma/\sqrt{r}$ and $\min_{r \in \{1, \ldots, n\}} U(r) := U(r^*)$ with $r^*$ being the smallest time-point where the equality holds. By using the Haar wavelet system in Algorithm 1, we have with probability at-least $1 - \delta$ that*

$$|\hat{\theta}_1 - \theta_1| \leq \kappa \cdot U(r^*),$$

*where $\kappa = (4\sqrt{2\log(\log n/\delta)} \vee 2\sqrt{2})(\log_2 n + 1)$ and $(a \vee b) := \max\{a, b\}$.*

Although the above theorem is a testament to the versatility of wavelet-based methods for the estimation problem, we remind the reader that it is only an upper-bound to the general bound developed in Lemma 1. Lemma 1 holds the potential to achieve faster estimation error rates by leveraging the sparsity of wavelet coefficients in any user-specified wavelet transformed space. This is indeed corroborated by

our numerical experiments in Section 6.1. Though we do not provide a specific construction where Lemma 1 can lead to faster error rate than that of Theorem 2, we empirically compare the corresponding upperbounds in Fig.5 (Appendix E) to validate this idea. However, a limitation of Lemma 1 is that for general wavelet transform matrices, it does not guarantee a rate that is at-least as good as the one given by Theorem 2, obtained via using Haar wavelets. In the following corollary, we derive a fail-safe guarantee obtained by using a specialized wavelet system, namely the CDJV wavelets (Cohen et al., 1993), in Algorithm 1.

**Corollary 3.** *For a sequence $\theta_{a:b}$ ($a > b$), define $TV(\theta_{a:b}) := \sum_{j=b+1}^{a} |\theta_j - \theta_{j-1}|$. For a timepoint $r$, let $S(r) := \{1, 2, 4, \ldots, 2^{\lfloor \log_2 r \rfloor}\}$. Let $\tilde{U}(r) := \max_{t \in S(r)} |TV(\theta_{t:1}) - \theta_1|) \vee \sigma/\sqrt{r}$ and $\min_{r \in \{1, \ldots, n\}} \tilde{U}(r) := \tilde{U}(r^*)$ with $r^*$ being the smallest time-point where the equality holds. Define $\kappa = (4\sqrt{2 \log(\log n/\delta)} \vee 2\sqrt{2})(\log_2 n + 1)$. By using CDJV wavelet transform with 2 vanishing moments in algorithm 1, we have with probability at-least $1 - \delta$ that*

$$|\hat{\theta}_1 - \theta_1| = O\left(\min\left\{\sum_{i \in \mathcal{I}} 6|W_{i,n}| \cdot (|\beta_i| \wedge \lambda), \kappa\tilde{U}(r^*)\right\}\right).$$

The above corollary guarantees a robust fail-safe estimation error. However, there are some caveats when comparing it against the result given in Theorem 2. Consider the scenario where the minimum in the bound given in Corollary 3 is achieved by $\tilde{U}(r^*)$ term. We remark that in such a situation, the bound in Theorem 2 can be tighter because $U(r^*) \leq \tilde{U}(r^*)$. However, as we shall see in Section 5, a bound of the form given in Corollary 3 is already sufficient to imply optimal estimation rates for the TV-denoising problem. This illustrates one utility of $\tilde{U}(r^*)$ type bounds.

## 3. Effect of Distribution Shift on the Model Performance

In the previous section, we proposed wavelet-based methods for estimating quantities during temporal distribution shifts. We now explore the rationale behind developing learning algorithms specifically designed for distribution shift scenarios. This section aims to theoretically demonstrate how and why model performance deteriorates when the training and testing distributions deviate. We first offer a theoretical examination of the machine learning model's loss function and comparing its behavior across training and testing datasets. Through our analysis, we establish upper and lower bounds for the loss function, which are directly influenced by the proportion of dissimilarly distributed/shifted samples. All proofs are deferred to the Appendices.

We first derive the log-loss of an ML model via its predictive distribution $p_\theta(y|\mathbf{x})$, with input covariates $\mathbf{x}$ and model output $y$, and testing data $D_{T_s}$ distribution, with $\theta$ denotes the model parameters. Note that $p_\theta(y|\mathbf{x})$ is obtained by training the model with training data $D_{T_r}$.

**Lemma 4.** *(Two Forms of the Log-Loss Function, adapted from (Achille and Soatto, 2018)) The log-loss $\mathcal{L}(\theta, D_{T_r}, D_{T_s})$ can be expressed in two forms*

$$\mathcal{L}(\theta, D_{T_r}, D_{T_s}) = KL\left(p_{D_{T_s}}(y|\mathbf{x})||p_\theta(y|\mathbf{x})\right) + \mathcal{H}\left(p_{D_{T_s}}(y|\mathbf{x})\right) \quad (1)$$

$$\mathcal{L}(\theta, D_{T_r}, D_{T_s}) = KL(p_{D_{T_s}}(\mathbf{x}, y)||p_\theta(\mathbf{x}, y)) - KL(p_{D_{T_s}}(\mathbf{x})||p_\theta(\mathbf{x})) + \mathcal{H}\left(p_{D_{T_s}}(y|\mathbf{x})\right) \quad (2)$$

*in which $p_{D_{T_s}}(y|\mathbf{x})$ and $p_\theta(y|\mathbf{x})$) are the conditional distribution of the label for testing data and predictive distribution, respectively; $KL\left(p||q\right)$ is the Kullback–Leibler divergence of two distributions $p$ and $q$; and $\mathcal{H}\left(p\right)$ is the entropy of a distribution $p$.*

During the model training, we only have access to $D_{T_r}$ and the loss function is measured/tested on a validation data which is a subset of $D_{T_r}$. Substituting $D_{T_r}$ into $D_{T_s}$ in Eq.(1), we have $\mathcal{L}_{training}(\theta, D_{T_r}) \sim KL(p_{D_{T_r}}(y|\mathbf{x})||p_\theta(y|\mathbf{x}))$ as $\mathcal{H}\left(p_{D_{T_s}}(y|\mathbf{x})\right)$ is a constant, which is independent of the ML model and its conditional probability/logit function $p_\theta(y|\mathbf{x})$. To achieve the minimum value for the log-loss, the KL-divergence should be driven towards 0.

As the focus is to study the effect of distribution shift, we will assume an infinite number of samples from the training data for perfect model training, and hence neglecting concentration arguments for the ease of presentation.

**Assumption 5.** *(Perfect ML Training) To investigate the effect of data distribution shift, we will assume a perfect ML model and training process, i.e. we are able to achieve $p_\theta(y|\mathbf{x}) \equiv p_{D_{T_r}}(y|\mathbf{x})$.*

Note that without Assumption 5, the model can train with $D_{T_r}$ distribution but output another distribution $\mathcal{D}$, and if $\mathcal{D} \equiv D_{T_s}$ then a seemingly bad model can perform surprisingly well. In subsequent sections, we will consider more realistic assumptions, i.e. limited training data and imperfect machine learning training processes.

From the first form of the log-loss function, i.e, Eq.(1), it is tempting to already conclude that distribution shift leads to model degradation. However, the first form Eq.(1) only refers to the divergence in the predictive distribution and conditional distribution of the label. In practical and real-world scenarios, it is more general and realistic to consider a shift in the dataset/joint probability $(\mathbf{x}, y)$ (Quionero-Candela et al., 2009; Harutyunyan et al., 2020). For example, when we add more samples into the training data, such addition

leads to a shift in $p_{D_{Tr}}(\mathbf{x}, y)$ first and foremost instead of directly leading to a shift in $p_{D_{Tr}}(y|\mathbf{x})$. The second form Eq.(2) provides the connection between the sample $(\mathbf{x}, y)$ and covariance $\mathbf{x}$ shift.

The first term in Eq.(1) and the first two terms in Eq.(2) represents the loss due to the distribution shift. That loss is minimizable by optimizing the model parameters $\theta$. The term $\mathcal{H}\left(p_{D_{Ts}}(y|\mathbf{x})\right)$ represents the intrinsic error (Achille and Soatto, 2018) to learn a dataset and it is not optimizable. For example, assume that we have an image dataset with two labels "dog" and "cat". For each image $\mathbf{x}$ we toss a coin and label the image as "dog" if head, and as "cat" if tail, irrespective of the image itself. In such case, $\mathcal{H}\left(p_{D_{Ts}}(y|\mathbf{x})\right)$ and the log-loss are very high (in fact, maximized) and the model performance is extremely bad, even if we have perfect ML training and use the test dataset to train the model.

To investigate the effect of distribution shift, we consider two scenarios usually observed in practice. Specifically, we define Training Distribution Shift Scenario as when the testing data is fixed, and we are able to gather more data samples in the past and train the model with new data to make better predictions. Conversely, we define Testing Distribution Shift Scenario when we collect the training data and train the model only once. Afterwards, the model is used for inference for all testing samples in the future.

**Definition 6.** *(Training (or Testing) Distribution Shift Scenario) The training (or testing) data distribution $D_T$ consists of two distributions $D_{T1}$ and $D_{T2}$. Denote $\alpha$ and $\beta$ as the proportion of samples from $D_{T1}$ and $\mathcal{D}_{T2}$ in $D_T$, respectively with $\alpha + \beta = 1$, we have*

$$p_{D_T}(\mathbf{x}, y) = \alpha p_{D_{T1}}(\mathbf{x}, y) + \beta p_{D_{T2}}(\mathbf{x}, y) \qquad (3)$$

*Here $T$ represents training data $T \equiv Tr$ (or testing data $T \equiv Ts$) and $T1 \equiv Ts$ (or $T \equiv Tr$).*

Under the Training Distribution Shift Scenario, the training data $T \equiv Tr$ consists of (1) samples from the testing distribution $T1 \equiv Ts$, and (2) samples from a dissimilar distribution $T2$. When, more and more dissimilarly-distributed samples from $T2$ are added into the training distribution, this leads to higher dissimilarity ratio $\beta$. Under the Testing Distribution Shift Scenario, the testing data $T \equiv Ts$ consists of (1) samples from the training distribution $T1 \equiv Tr$, and (2) samples from a dissimilar distribution $T2$. When, more and more dissimilarly-distributed samples from $T2$ are added into the testing distribution, this leads to higher dissimilarity ratio $\beta$.

We are now ready to investigate the effect of the distribution shift. One of our contributions is the following theorem

**Theorem 7.** *Under Assumption 5 and for distribution shift scenarios specified in Definition 6, the log-loss is bounded above and below by monotonic non-decreasing functions on $\beta$: $LB(\beta) \leq \mathcal{L}(\theta, D_{Tr}, D_{Ts}) \leq UB(\beta)$.*

A direct consequence of Theorem 7 is the below corollary

**Corollary 8.** *Under Assumption 5, the model performs worse when we add more samples from a distribution, which is different from the testing data, into the training data. Conversely, the model performs better when we add more samples from the testing distribution into the training data.*

Theorem 7, established under Assumption 5 (which assumes an idealized machine learning model and training process), provides best-case performance boundaries. The lower bound thus reveals an inherent limitation: regardless of the sophistication of our learning algorithm, model performance will decline as the dissimilarity ratio $\beta$ increases. Despite recent literature suggesting that doing naive ERM might perform adequately across different distributions, our analysis demonstrates that predictive power will substantially deteriorate in worst case, as training and testing dsitribution diverge. It is worth noting that we need both lower and upper bounds of the log-loss. The reason is that the lower bound in Theorem 7 can only says how worse the loss can become. The further from 0, the worse the loss is. When $\beta$, the lower bound reduces to the conclusion that the loss is larger than 0 (see Appendix D), which is trivial. Only an upper-bound confirms that the model gets better when having more similarly distributed training samples versus the testing data.

## 4. Learning Under Temporal Distribution Shift

In the previous section, our analysis reveals the performance decline inherent when training and testing distributions differ. Consequently, developing robust learning algorithms that can mitigate this degradation is an important research objective. In this section, we study the fundamental statistical problem of binary classification under training distribution shift. We are given access to $n$ covariate-target pairs: $(\mathbf{x}_n, y_n), \ldots, (\mathbf{x}_2, y_2), (\mathbf{x}_1, y_1)$ along with a hypothesis class $\mathcal{F}$. We assume that for each $i \in [1, \ldots, n]$, $(\mathbf{x}_i, y_i)$ is independently sampled from a distribution $D_i$. Let $\ell(f(\mathbf{x}), y) := \mathbb{I}\{f(\mathbf{x}) \neq y\}$ be the loss associated with a hypothesis $f \in \mathcal{F}$ on the example $(\mathbf{x}, y)$ where $\mathbb{I}$ is the binary indicator function. The population level loss of a hypothesis $f$ on the testing distribution $D_1$ is $L_f := E_{(\mathbf{X}, Y) \sim D_1}[\ell(f(\mathbf{X}), Y)]$. We are interested in finding a hypothesis whose population level loss on testing Distribution $D_1$ is close to $\min_{f \in \mathcal{F}} L_f := L_{f_*}$ for some $f_* \in \mathcal{F}$. More precisely we are interested in finding a hypothesis $\hat{f}$ whose excess risk given by $L_{\hat{f}} - L_{f_*}$ is controlled.

This differs from standard statistical learning, where we assume $n$ i.i.d. samples from $D_1$ are used to learn a hypothesis (Bousquet et al., 2004). Since we observe only a single

labelled datapoint from $D_1$, pooling recent data based on shift severity is crucial. The challenge is determining how much past data to use for training, without prior insight into the shift's intensity. To address this, we perform ERM using the loss estimates from Section 2, leading to the following excess risk control.

**Theorem 9.** *For a function $f \in \mathcal{F}$, we defined obtain its estimated loss $\hat{\ell}_f$ as follows. Run Algorithm 1 with data $\ell(f(\mathbf{x}_n), y_n), \ldots, \ell(f(\mathbf{x}_1), y_1)$ as input. Let $\hat{\ell}_f$ be the estimate returned by Algorithm 1. The ERM is defined by $\hat{f} \in argmin_{f \in \mathcal{F}} \hat{\ell}_f$.*

*For a function $f \in \mathcal{F}$, let $\boldsymbol{\theta}^f := [E[l(f(\mathbf{X}_n), y_n)], \ldots, E[l(f(\mathbf{X}_1), y_1)]]^T$. Let $\boldsymbol{\beta}^f := \boldsymbol{W}\boldsymbol{\theta}^f$ for a wavelet transform matrix $\boldsymbol{W}$. Consider the index set $\mathcal{I}$ defined in Lemma 1. Let $d$ be the VC dimension of $\mathcal{F}$. We have with probability at-least $1 - \delta$,*

$$L_{\hat{f}} - L_{f_*} \le 8\sqrt{\frac{2d\log(2n)}{n}} + 2\sqrt{\frac{2\log(3/\delta)}{n}}$$
$$+ \sqrt{3} \cdot \sup_{f \in \mathcal{F}} \sum_{i \in \mathcal{I}} 6|W_{i,n}| \cdot (|\beta_i^f| \wedge 2\sigma\sqrt{4d\log(3\log 2n/\delta)}).$$

Assuming that the wavelet transform in Theorem 9 is the Haar transform and applying Theorem 2, we obtain the following corollary.

**Corollary 10.** *Let $\bar{D}_t = \frac{1}{t} \sum_{i=1}^{T} D_i$. Define $\|\bar{D}_t - D_1\|_{\mathcal{F}} := \sup_{f \in \mathcal{F}} |E_{(X,Y)\sim\bar{D}_t}[\ell(f(X), Y)] - E_{(X,Y)\in D_1}[\ell(f(X), Y)]|$ as the discrepancy measure. Consider $\boldsymbol{W}$ in Theorem 9 to be the Haar transform matrix. Then the excess risk is bounded with probability at-least $1 - \delta$ by*

$$L_{\hat{f}} - L_{f_*} \le 8\sqrt{\frac{2d\log(2n)}{n}} + 2\sqrt{\frac{2\log(3/\delta)}{n}} +$$
$$\kappa \cdot \min_{r \in [n]} \left( \max_{t \in S(r)} \|\bar{D}_t - D_1\|_{\mathcal{F}} \vee \sqrt{d/r} \right),$$

*where $\kappa = (4\sqrt{4\log(3\log 2n/\delta)} \vee 2\sqrt{2})(\log_2 n + 1)$ and $S(r)$ is as defined in Theorem 2.*

**Discussion of our results in contrast to prior work.** We remark that an optimal answer to the problem of classification under temporal training distribution shift has been given by Mazzetto and Upfal (2023) in the form of a bound similar to that in Corollary 10. Their solution relies on making $O(\log n)$ calls to an ERM oracle to obtain a minimizer that can control the excess risk under the distribution $D_1$. In contrast, our solution requires *exactly one* call to the ERM oracle. The tradeoff for this computational improvement is the inflation of the excess risk bound of the prior work only by a factor of $O(\log n)$. We remark that the result in Theorem 9 is more general than the prior work since it connects

the excess risk to sparsity (and hence the degree of temporal stationarity) of the sequence of population level losses in a wavelet transformed domain. Hence this bound can be potentially much sharper than those obtained by constraining the wavelet transform to be Haar. A fail-safe guarantee similar to that of Corollary 3 can be obtained by using CDJV wavelets.

We note that the logarithmic improvement in computation stems from a simple but careful technical observation. More precisely, one can reduce the computational complexity of the procedure in Mazzetto and Upfal (2023) by estimating the loss of each function $f \in \mathcal{F}$ separately while performing the ERM. This is realizable by taking the input function class to their algorithm to be the singleton set $\{\ell_f\}$ with $\ell_f$ defined by the mapping $\ell_f : \mathcal{X} \times \mathcal{Y} \to \ell(f(\mathbf{x}), y)$. However, doing so poses a major challenge when optimizing with surrogate losses. We elaborate on this aspect as follows.

Though the result in Theorem 9 concerns with binary zero-one loss, one is often interested in solving the ERM (possibly sub-optimally) via the use of differentiable surrogate losses and optimizing the objective via gradient descent. Given a differentiable surrogate loss, one can form an ERM objective based on soft-thresholding the surrogate loss sequence similar to what is done in Theorem 9. Such an objective is differentiable almost everywhere. In contrast, estimating the loss of each function separately using the procedure from Mazzetto and Upfal (2023) renders the ERM objective non-differentiable. The non-differentiability stems from the fact that their estimator is constructed based on the output of a sequence of data-dependent binary comparisons.

## 5. Optimal Error Rates for Total-Variation Denoising

In this section, we prove a black-box result that *any* algorithm that satisfies bounds of the type studied in Theorem 2 or Corollary 3 can imply optimal estimation error rates for TV-denoising problem. To the best of our knowledge, such an implication has been not uncovered before in literature. Apart from being a result of theoretical interest, it helps uncover the previously unknown optimality of some existing algorithms for the problem of TV-denoising.

In TV-denoising problem, we are given access to $n$ observations of the form $y_t = \theta_t + \epsilon_t, \quad t = 1, \ldots, n$ where $\epsilon_t$ are i.i.d. $\sigma$-sub-gaussian observations. Our goal is to form estimates $\hat{\theta}_{1:n}$ of the groundtruth $\theta_{1:n}$ from the noisy observations. To impose regularity/structure to the underlying ground truth, one can define the Total Variation (TV) class as follows.

$$\mathcal{TV}(C) = \left\{ \theta_{1:n} : \sum_{t=2}^{n} |\theta_t - \theta_{t-1}| \le C \right\},$$

wherein the quantity $C$ is viewed as the *radius* of the TV class. We assume that the groundtruth sequence satsifies $\theta_{1:n} \in \mathcal{TV}(C)$. We remark that one can also define alternate sequence classes based on penalising the sum of squared differences of a sequence. For eg. $\sum_{t=2}^{n} |\theta_t - \theta_{t-1}|^2 \le \tilde{C}^2$. We defer the reader to Baby and Wang (2019) for an exposition on why the TV class is more expressive and statistically challenging to estimate than such sequence classes.

To measure the quality of risk estimates, we focus on two commonly used risk functionals in practice as follows: $R_{\text{sq}}(\hat{\theta}_{1:n}, \theta_{1:n}) = \sum_{t=1}^{n} (\hat{\theta}_t - \theta_t)^2$ and $R_{\text{abs}}(\hat{\theta}_{1:n}, \theta_{1:n}) = \sum_{t=1}^{n} |\hat{\theta}_t - \theta_t|$.

We are interested in coming up with estimators that can attain this min-max error lower-bounds (upto log terms).

**Theorem 11.** *Suppose that an algorithm satisfies a bound of the form given in Theorem 2 or Corollary 3. Consider running the algorithm iteratively to produce the estimates $\hat{\theta}_{1:n}$. Then with probability at-least $1 - \delta$ we have that*

$$R_{sq}(\hat{\theta}_{1:n}, \theta_{1:n}) = \tilde{O}(n^{1/3} C^{2/3} \sigma^{4/3})$$
$$R_{abs}(\hat{\theta}_{1:n}, \theta_{1:n}) = \tilde{O}(n^{2/3} C^{1/3} \sigma^{2/3}).$$

These rates can be shown to be optimal modulo log terms (Donoho et al., 1996). We note that the algorithms in Mazzetto and Upfal (2023); Han et al. (2024) already satisfy a bound of the form in Theorem 2. Thus Theorem 11 uncovers new off-the-shelf optimal algorithms for the TV-denoising problem.

## 6. Experimental Results

In this section, we validate our findings on synthetic and real-world data.

### 6.1. Experiments on Synthetic Data

In this section, we report the results obtained from simulation studies. As discussed before, the algorithm from Mazzetto and Upfal (2023) aims to capture the non-stationarity in terms of local averages of a sequence. This can potentially over-smooth/under-smooth trends in the ground-truth which may not always assume a piece-wise constant structure. In this section, we show that Algorithm 1 is superior in capturing a wider range of trend patterns, while Lemma 1 and Theorem 2 achieves faster estimation error rates.

**Baselines.** 1) DB8 is the version of Algorithm 1 that uses Daubechies wavelet with 8 vanishing moments. 2) HAAR

is the version of Algorithm 1 that uses Haar wavelets. 3) AVG is the algorithm from Mazzetto and Upfal (2023). 4) ARW is the algorithm from Han et al. (2024). 5) Aligator is the trend forecasting algorithm (hedged version) from Baby et al. (2021).

**Experimental methodology.** First a ground-truth signal is generated. We considered two types of ground truth signal as shown in Fig.4. The Random signal features a signal which can be either 0 or 1 based on the output a fair coin flip and Doppler features a trend with spatially varying degree of smoothness. All signals lie within the range $[-1.5, 1.5]$. The ground-truth sequence is generated by sampling these signals at 500 equispaced points. To generate the observations, we add independent uniformly sampled noise from $[-B, B]$ to the ground-truth signal for each of the 500 time-points. The quantity $B$ is varied across the levels $\{0.2, 0.3, 0.5, 0.7, 1\}$ to study the behaviour of the algorithms across different noise-levels. At any time-point, the baselines (except Aligator) use all the observations from history including the current time-point to estimate the ground-truth. The Mean Squared Error (MSE) across all 500 time-points are measured. All experiments were conducted across 5 trials. We report the average MSE and associated standard deviations across the trials. The failure probability parameter for all algorithms is set to be 0.1.

We report the results by considering two scenarios. In the first case, the standard deviation of the noise is exactly known. In this case we adjust the constants in Algorithms 1 and that of AVG according to their theoretically optimal values. Next, we consider the case of no prior knowledge of the standard deviation. The algorithms ARW and Aligator are directly applicable to this scenario since they do not require prior knowledge of the standard deviation. As a consequence, for a fair comparison, we report another table in which AVG and wavelet based thresholding algorithms are executed via bounding or estimating the standard deviation while displaying alongside the results from algorithms ARW and Aligator. For the wavelet-based algorithms, an estimate of the standard deviation is formed based on the Median Absolute Deviation (MAD) of the wavelet coefficients at the highest resolution similar to as done in Donoho et al. (1998). For AVG, a bound on the observed values are used as a proxy. The results are reported for the known standard deviation and the case of unknown deviation respectively in Tables 1 and 2.

**Interpretation of results.** For the case of known standard deviation, the wavelet-based algorithms are found to perform better than AVG. In almost all cases, we see significant improvements via the usage of wavelets across various noise levels with biggest improvements observed in the high signal-noise-ratio regime. This validates the ability of wavelets to produce sharp estimation error rates. In particu-

| Noise Level | AVG | HAAR (*This paper*) | DB8 (*This paper*) | | Noise Level | AVG | HAAR (*This paper*) | DB8 (*This paper*) |
|---|---|---|---|---|---|---|---|---|
| 0.2 | $0.205 \pm 0.0041$ | $0.094 \pm 0.0014$ | $\mathbf{0.0661 \pm 0.0026}$ | | 0.2 | $0.511 \pm 0.0013$ | $0.068 \pm 0.0035$ | $\mathbf{0.007 \pm 0.0002}$ |
| 0.3 | $0.202 \pm 0.0098$ | $0.154 \pm 0.0087$ | $\mathbf{0.091 \pm 0.0040}$ | | 0.3 | $0.510 \pm 0.0020$ | $0.103 \pm 0.0060$ | $\mathbf{0.013 \pm 0.0003}$ |
| 0.5 | $0.209 \pm 0.0077$ | $0.191 \pm 0.0065$ | $\mathbf{0.119 \pm 0.0042}$ | | 0.5 | $0.509 \pm 0.0035$ | $0.155 \pm 0.0113$ | $\mathbf{0.032 \pm 0.0007}$ |
| 0.7 | $0.208 \pm 0.0082$ | $0.182 \pm 0.0077$ | $\mathbf{0.147 \pm 0.0051}$ | | 0.7 | $0.508 \pm 0.0052$ | $0.155 \pm 0.0162$ | $\mathbf{0.061 \pm 0.0014}$ |
| 1 | $0.214 \pm 0.0049$ | $\mathbf{0.174 \pm 0.0100}$ | $0.212 \pm 0.0062$ | | 1 | $0.508 \pm 0.0081$ | $0.155 \pm 0.0224$ | $\mathbf{0.129 \pm 0.0028}$ |

(a) Ground truth: Random              (b) Ground truth: Doppler

Table 1: MSE of different algorithms when the noise standard deviation is known. Groundtruth signals used are shown in Fig.4 (Appendix E).

| Noise Level | AVG | ARW | Aligator | HAAR (*This paper*) | DB8 (*This paper*) |
|---|---|---|---|---|---|
| 0.2 | $0.210 \pm 0.0045$ | $0.203 \pm 0.0079$ | $0.214 \pm 0.0086$ | $0.243 \pm 0.0218$ | $\mathbf{0.188 \pm 0.0155}$ |
| 0.3 | $0.204 \pm 0.0060$ | $0.205 \pm 0.0079$ | $0.214 \pm 0.0071$ | $0.258 \pm 0.0224$ | $\mathbf{0.188 \pm 0.0191}$ |
| 0.5 | $\mathbf{0.210 \pm 0.0080}$ | $0.214 \pm 0.0078$ | $0.212 \pm 0.0066$ | $0.271 \pm 0.0087$ | $0.214 \pm 0.0058$ |
| 0.7 | $\mathbf{0.209 \pm 0.0081}$ | $0.205 \pm 0.0053$ | $0.222 \pm 0.0070$ | $0.270 \pm 0.0081$ | $0.238 \pm 0.0143$ |
| 1 | $0.219 \pm 0.0062$ | $\mathbf{0.211 \pm 0.0115}$ | $0.213 \pm 0.0103$ | $0.289 \pm 0.0196$ | $0.307 \pm 0.0151$ |

(a) Ground truth: Random

| Noise Level | AVG | ARW | Aligator | HAAR (*This paper*) | DB8 (*This paper*) |
|---|---|---|---|---|---|
| 0.2 | $0.512 \pm 0.0021$ | $0.3844 \pm 0.0052$ | $0.234 \pm 0.0021$ | $0.053 \pm 0.0017$ | $\mathbf{0.0204 \pm 0.0007}$ |
| 0.3 | $0.512 \pm 0.0032$ | $0.389 \pm 0.0077$ | $0.234 \pm 0.0032$ | $0.056 \pm 0.0018$ | $\mathbf{0.0265 \pm 0.0010}$ |
| 0.5 | $0.512 \pm 0.0053$ | $0.400 \pm 0.0136$ | $0.235 \pm 0.0055$ | $0.065 \pm 0.0018$ | $\mathbf{0.0444 \pm 0.0016}$ |
| 0.7 | $0.512 \pm 0.0074$ | $0.402 \pm 0.0181$ | $0.235 \pm 0.0078$ | $0.072 \pm 0.0031$ | $\mathbf{0.070 \pm 0.0021}$ |
| 1 | $0.514 \pm 0.0106$ | $0.407 \pm 0.0215$ | $0.235 \pm 0.0111$ | $\mathbf{0.088 \pm 0.0035}$ | $0.129 \pm 0.0058$ |

(b) Ground truth: Doppler

Table 2: MSE of different algorithms when the noise standard deviation is unknown.

lar, the usage of higher order DB8 wavelet produces lowest estimation error in most cases since the DB8 wavelet system can capture complex trends more effectively than the HAAR system (which is also known as the DB1 wavelet which is of lower order than DB8). Similar observations are carried forward to the unknown standard deviation regime. Both Aligator and wavelet based methods outperforms ARW and AVG in most cases. A fundamental difference between these algorithms is that ARW and AVG are based on the estimate produced by a single predictor, while Aligator and wavelet based algorithms ensembles a set of base predictors to form the final prediction. For wavelet based methods, the base predictors are the wavelet bases and ensemble weights are precisely the denoised wavelet coefficients. Intuitively, the ensemble nature of these algorithms enables to outperform the counterparts by achieving an improved variance reduction. We remark that we used theoretically recommended values for constants that guide the restart/selection decisions in AVG and ARW. Tuning those constants in an online setting is otherwise unclear. The algorithm Aligator is known to produce minimax optimal MSE rates for

estimating a ground truth signal from noisy observations (Baby et al., 2021). However, a point-wise bound on the estimation quality similar to Theorem 2 is not known for Aligator. In contrast, the methods HAAR and DB8 comes with strong point-wise error bounds (Lemma 1 and Theorem 2) which translates to improved performance in practice. In the unknown standard deviation regime, we see that AVG and ARW performs slightly better than HAAR and DB8 for certain noise levels for the Random signal. AVG and ARW are designed to detect (nearly) stationary portions in a signal and average the stable portions to compute the final estimates. Hence Random signal by virtue of its piecewise constant nature becomes a favourable prior to AVG and ARW performing slightly better than other methods with when provided with noisy estimates of the standard deviation. However, in almost all cases for Random signal, the performance of all algorithms are roughly comparable (with difference only in the second significant digit, see Table 2) which in turn emphasizes the robustness of ensemble based methods like Aligator, HAAR and DB8 even when the standard deviation is unknown. Further empirical results

| Method | 79%-1% Split | 75%-5% Split |
|---|---|---|
| ARW | $0.0790 \pm 0.0005$ | **0.0719±0.0005** |
| HAAR (ours) | **0.0722±0.0006** | $0.0736 \pm 0.0011$ |
| DB8 (ours) | $0.0762 \pm 0.0002$ | $0.0768 \pm 0.0008$ |

Table 3: Average test MSE $\pm$ standard error on the Dubai Land Department dataset over 192 months and 5 runs.

are deferred to Appendix E.

### 6.2. Experiments on Real Data

As an application of our proposed methods, we conduct a model selection experiment using real-world data. We evaluate our method on data from the Dubai Land Department (Land Sales) following the setup identical to that of [2]. The dataset includes apartment sales from January 2008 to December 2023 (192 months). Each month is treated as a time period, where the goal is to predict final prices based on apartment features. Data is randomly split into 20% test, with two train-validation splits: (a) $79\% - 1\%$ and (b) $75\% - 5\%$.

For each month $t$, we train Random Forest (Breiman, 2001) and XGBoost (Chen and Guestrin, 2016) models using a window of past data where we consider window sizes $w \in [1, 4, 16, 62, 256]$, yielding 10 models per month. Validation mean squared error (MSE) from past and current months are used to refine the current month's estimate of MSE via Algorithm 1 or ARW from Han et al. (2024). The refined validation scores are used to select the best model for final MSE evaluation on test data. We report the average MSE of this model selection scheme over 192 months and 5 independent runs, comparing our method to ARW. In Table 3, HAAR and DB8 are versions of Algorithm 1 with the corresponding wavelet basis given as input.

We see that wavelet based methods shine especially when the validation data is scarce. This allows us to include more data for training while still allowing to obtain high quality estimates for validation scores. Such a property can be especially helpful in data-scarce regimes. Unlike the synthetic data experiments, here we find that Haar wavelets perform better than DB8. This can be attributed to the following facts: i) the noise in the observations depart from iid sub-gaussian assumption and; ii) the high degree of non-stationarity in the pricing data as indicated by Fig.2 makes the underlying trends to have a low degree piecewise polynomial structure. This render the groundtruth irregular (or less smooth) which can be suitably handled by lower order Haar wavelets which are also less smooth and abrupt (see Fig.3).

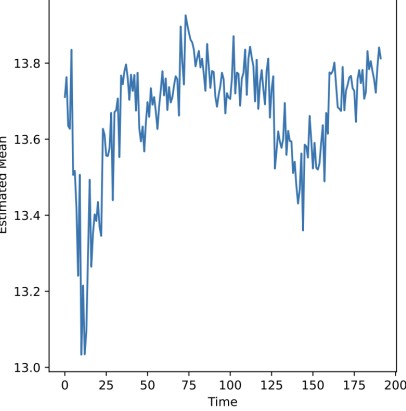

Figure 2: Mean house prices vs time in Dubai housing data (reproduced from Han et al. (2024)). The pricing trend is abrupt and highly non-stationary.

### 6.3. Notes to Practitioners about Wavelet Selection

We close off the experimental section by few remarks about how one can select a wavelet system in practice for the use in Algorithm 1. Practitioners typically analyze data trends—if they follow a piecewise polynomial of degree $k$, a wavelet of order $k + 1$ is chosen, as it effectively models such structures. However, higher-order wavelets introduce numerical instabilities and variance, though the latter can be mitigated with more data. Selecting a wavelet basis is akin to choosing a kernel in Gaussian Processes—application-specific and guided by practical intuition. A more comprehensive exposition can be obtained from Mallat (2008).

## 7. Conclusion, Limitations and Future Work

We introduced a wavelet-based approach for estimating non-stationary time series with strong point-wise error guarantees. Our solution offers a fairly general perspective to this problem compared to prior works. Using Haar wavelets recovers prior results, while advanced wavelets enable estimation of complex trends. Our findings extend to binary classification, leading to efficient algorithms, and uncover new optimal methods for TV-denoising. Experiments on real and synthetic data show notable improvements over previous work.

There are some limitations that we leave to be addressed in future works. Principled ways to select an appropriate wavelet system for a given task of interest needs additional investigation. Unlike prior approaches that use adaptive window sizes for moving averages, our method implicitly leverages relevant data portions. However, maintaining a window-size can have meaningful implications to change-point detection. Exploring the utility of our methods to this extension remains as an interesting direction to explore.

## Acknowledgments

We thank Jiaxing Zhang and Alessio Mazzetto for insightful early discussions that helped shape the direction of this work.

## Impact Statement

This paper presents work whose goal is to advance the field of Machine Learning. There are many potential societal consequences of our work, none which we feel must be specifically highlighted here.

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

# A. Related Work

**Effect of Distribution Shift** It is commonly believed that when the training data differs from the testing data, the model's performance will degrade (Quionero-Candela et al., 2009; Harutyunyan et al., 2020; Wang and Mao, 2023; Zhang et al., 2022; Federici et al., 2021; Achille and Soatto, 2018; Cai et al., 2022). Several studies attempted to connect the distance of the training and testing distributions with the expected errors (Long, 1998; Ben-David et al., 2010; Mazzetto and Upfal, 2023), by upper-bounding the expected errors for certain function classes. In contrast, we analyze the model loss function directly and provide both the upper- and lower-bounds, thus concluding once and for all that the model performance degrades as the misalignment between training and testing data increases, and vice versa.

**Rolling Window Based Estimation Techniques.** As mentioned in Section 1, a natural way to balance the bias-variance trade-off in estimating $\theta_1$ is to use data from a carefully selected, *but then fixed*, window of most recent observations. Works such as (Hanneke et al., 2015; Mohri and Muñoz Medina, 2012; Bifet and Gavaldà, 2007; Mazzetto and Upfal, 2023; Han et al., 2024) follow this direction. In contrast, our algorithm, based on wavelet-denoising, does not explicitly maintain a window size, yet achieve optimal estimation error rates.

**Non-Stationary Online Learning.** There are various studies from non-stationary online learning that aim to track a moving target of ground-truth (Zinkevich, 2003; Zhang et al., 2018; Cutkosky, 2020; Besbes et al., 2015; Jadbabaie et al., 2015; Yang et al., 2016; Mokhtari et al., 2016; Chen et al., 2018; Baby and Wang, 2019; Zhao et al., 2020; Daniely et al., 2015; Jun et al., 2017; Hazan and Seshadhri, 2007; Baby and Wang, 2021; 2022; 2020; 2023; Baby et al., 2023; 2025). These works aim to estimate a moving target while controlling a cumulative performance metric such as MSE or (dynamic) regret. To the best of our knowledge, none of these works lead to point-wise estimation error guarantees which are strictly stronger than controlling cumulative performance metrics. We remark that our methods can also be used as a sub-routine for the online estimation problem by iteratively executing our algorithm for each round.

**Total-Variation Denoising.** The field of TV denoising studies the problem of estimating a ground truth sequence from noisy observations. Unlike the fixed-dimensional estimation procedures, no parametric structure is imposed, e.g., linear w.r.t. some covariates. Instead, the ground-truth is assumed to belong to an abstract class of sequences whose total variation is bounded by some unknown value. There are several algorithms that are known to be minimax optimal for this task (Mammen and van de Geer, 1997; Kim et al., 2009; Tibshirani, 2014b; Wang et al., 2015; Guntuboyina et al., 2017). In this paper we prove a fairly general result that any algorithm that satisfies a bound of the form given in Theorem 2 is by default an optimal algorithm for the TV-denoising problem.

# B. Preliminaries

In this section, we give a brief introduction to wavelets. We focus only on some details that are relevant to the discussion in this paper and refer the readers to Mallat (2008); Johnstone (2017) for a comprehensive overview.

## B.1. Discrete Haar Wavelet Transform

The discrete Haar wavelet transform (HWT) is a fundamental method in signal processing, designed to decompose a signal into components that represent its average and detailed variations. It operates efficiently by leveraging the simplicity of the Haar basis functions, making it particularly suitable for computational tasks.

### B.1.1. Coefficients in the Haar Wavelet Transform

The HWT splits a signal into one *approximation coefficient* and multiple *detail coefficients*. These coefficients are computed using the *Haar transform matrix $H$*, which operates on a signal $x \in \mathbb{R}^n$, where $n$ is a power of 2.

- **Approximation Coefficient:** The *first row* of the Haar transform matrix computes the approximation coefficient, which represents the global average (low-frequency content) of the signal. This coefficient aggregates information across the entire signal, providing a coarse summary.

- **Detail Coefficients:** The *remaining rows* of the Haar transform matrix compute the detail coefficients, which capture the differences (high-frequency content) at progressively finer scales. These coefficients reveal variations within smaller and smaller segments of the signal.

The rows of the matrix represents (discrete) wavelets. As an example, for a signal of length $n = 8$, the orthonormal Haar transform matrix $H$ is:

$$
H = \begin{bmatrix}
\frac{1}{\sqrt{8}} & \frac{1}{\sqrt{8}} & \frac{1}{\sqrt{8}} & \frac{1}{\sqrt{8}} & \frac{1}{\sqrt{8}} & \frac{1}{\sqrt{8}} & \frac{1}{\sqrt{8}} & \frac{1}{\sqrt{8}} \\
\frac{1}{\sqrt{8}} & \frac{1}{\sqrt{8}} & \frac{1}{\sqrt{8}} & \frac{1}{\sqrt{8}} & -\frac{1}{\sqrt{8}} & -\frac{1}{\sqrt{8}} & -\frac{1}{\sqrt{8}} & -\frac{1}{\sqrt{8}} \\
\frac{1}{\sqrt{4}} & \frac{1}{\sqrt{4}} & -\frac{1}{\sqrt{4}} & -\frac{1}{\sqrt{4}} & 0 & 0 & 0 & 0 \\
0 & 0 & 0 & 0 & \frac{1}{\sqrt{4}} & \frac{1}{\sqrt{4}} & -\frac{1}{\sqrt{4}} & -\frac{1}{\sqrt{4}} \\
\frac{1}{\sqrt{2}} & -\frac{1}{\sqrt{2}} & 0 & 0 & 0 & 0 & 0 & 0 \\
0 & 0 & \frac{1}{\sqrt{2}} & -\frac{1}{\sqrt{2}} & 0 & 0 & 0 & 0 \\
0 & 0 & 0 & 0 & \frac{1}{\sqrt{2}} & -\frac{1}{\sqrt{2}} & 0 & 0 \\
0 & 0 & 0 & 0 & 0 & 0 & \frac{1}{\sqrt{2}} & -\frac{1}{\sqrt{2}}
\end{bmatrix}.
$$

- The *first row* computes the approximation coefficient.

- All other rows compute differences between groups of values at various levels of resolution (detail coefficients).

### B.1.2. MULTI-RESOLUTION REPRESENTATION

The HWT follows a multi-resolution scheme where the detail coefficients are computed at various scales, progressively halving the resolution at each step. This hierarchical representation enables the efficient analysis and reconstruction of signals, capturing both coarse and fine features.

For a signal of length $n$, we can decompose the detail wavelets into $\log_2 n$ resolution/levels. At a reslution $j \in \{0, \ldots, \log_2 n - 1\}$, there are $2^j$ wavelets. These detail wavelets can be thought of obtained by scaling and translation of a mother wavelet. For Haar transform, the mother wavelet assumes the form

$$
\psi(t) = \begin{cases} 1, & 0 < t \leq n/2, \\ -1, & n/2 < t \leq n, \\ 0, & \text{otherwise.} \end{cases}
$$

where $t \in \{1, \ldots, n\}$. Detail wavelets at resolution $j$ are obtained as follows:

$$
\psi_{j,k}(t) = \sqrt{\frac{2^j}{n}} \psi(2^j t - nk), \ k \in \{0, \ldots, 2^j - 1\}.
$$

Consequently one can use a dyadic indexing scheme to refer to the associated wavelet coefficients. We can see that any wavelet at resolution $j$ has a support length of $n/2^j$. Further, different wavelets at same resolution have disjoint support. Due to these properties one can see that when we decompose the signal $x \in \mathbb{R}^n$ using the wavelet basis, only $\log_2 n + 1$ wavelets affects the value of the signal at any index $i$ which is $x[i]$. Here the count is $\log_2 n + 1$ because of the inclusion of the approximation wavelet as well (the first row of the HWT matrix).

### B.2. Other Wavelet Systems

While the discrete Haar wavelet transform (HWT) provides a foundational introduction to wavelet analysis, it is one specific case within the broader family of wavelet transforms. Many wavelet transforms have been developed to address limitations of the Haar transform, such as its lack of smoothness and inability to capture higher-order variations or trends in signals. These alternative wavelets offer enhanced smoothness, compact support, and better time-frequency localization.

### B.2.1. DAUBECHIES WAVELETS

The Daubechies wavelets, introduced by Ingrid Daubechies, are a prominent family of compactly supported orthogonal wavelets. These wavelets generalize the Haar transform by using smoother basis functions, making them ideal for

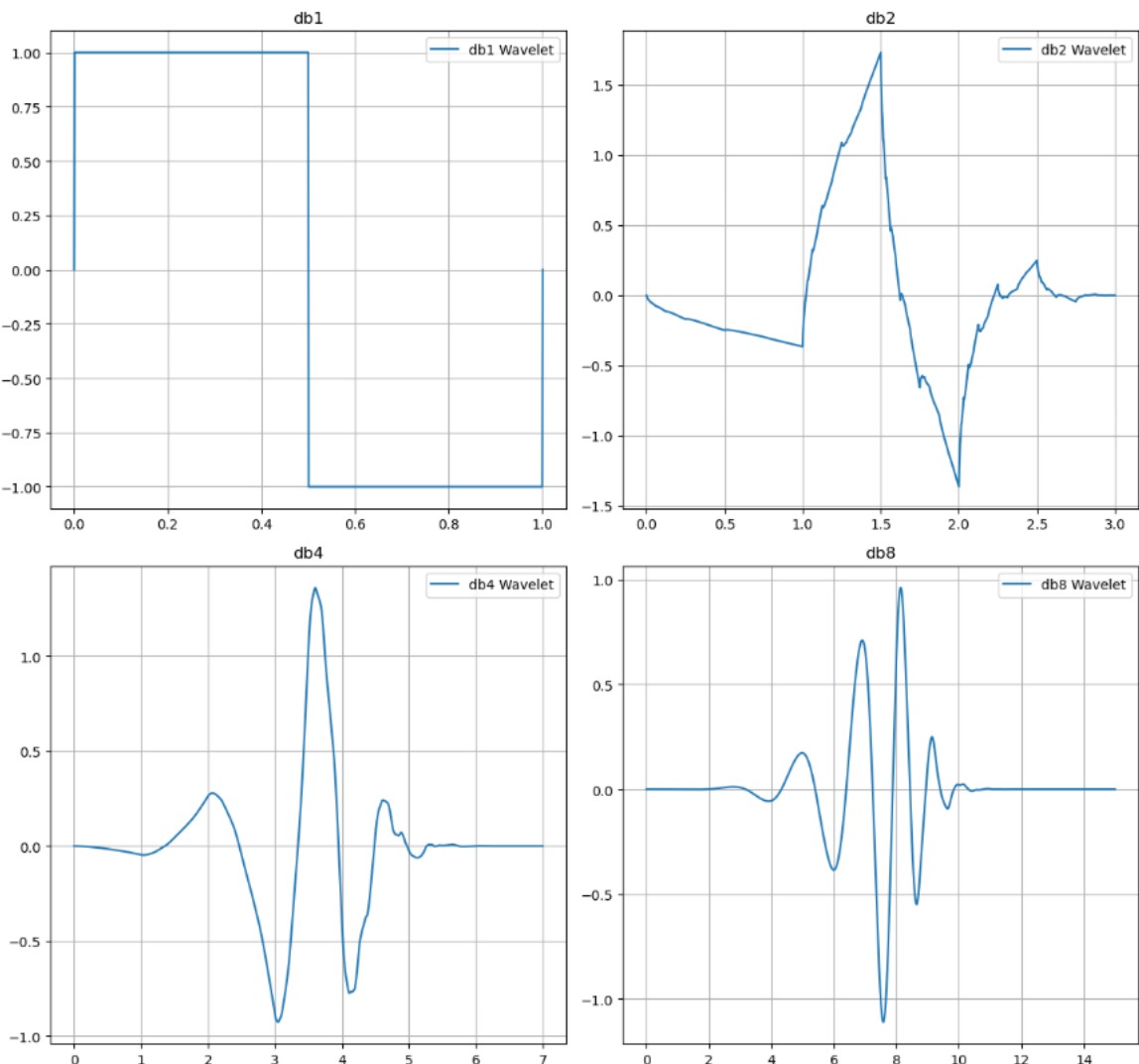

Figure 3: *Daubechies (DB) wavelets with increasing number of vanishing moments. We can see that the Haar system is a special case of the DB system with 1 vanishing moment. As we increase the number of vanishing moments, the wavelets get smoother.*

representing signals with continuous or slowly varying features. Each wavelet in the family is denoted as DBN where N is the number of vanishing moments. Higher-order Daubechies wavelets (DB2, DB4, etc.) offer better smoothness, which is advantageous in applications such as image processing and audio compression. The Haar wavelets is precisely the db1 system.

### B.2.2. CDJV WAVELETS

A special wavelet construction scheme based on Daubechies wavelets was proposed in Cohen et al. (1993). These wavelet functions enjoy near or exact symmetry in contrast to the Daubechies wavelets. The CDJV system is particularly useful in theoretical analysis due to equivalence between functional norms that capture smoothness of functions and Besov norms of the associated wavelet coefficients. A remarkable property is that the Total variation of a function can be completely represented using weighted sum of L1 norm (with weights larger than 1) of its associated CDJV wavelet coefficients at each resolution. We refer the reader to Donoho et al. (1998); Johnstone (2017) for an elaborate treatment. A high level overview of the construction of CDJV wavelets can be found in Qian et al. (2024).

## C. Proof of Lemma 4

We adapt (Achille and Soatto, 2018)[Proposition C.2] to prove the first form of the log-likelihood function. The original (Achille and Soatto, 2018)[Proposition C.2] lacks clarity in differentiating between $D_{Tr}$ and $p_\theta(y|\mathbf{x})$. Through our revision, we can articulate Assumption 5 with greater precision.

We have

$$
\begin{aligned}
\mathcal{L}(\theta, D_{Tr}, D_{Ts}) &= -\frac{1}{K} \log \prod_{(\mathbf{x}_k, y_k) \in D_{Ts}} p_\theta(Y = y_k|\mathbf{x}_k) = -\frac{1}{K} \log \prod_{(\mathbf{x}_k, y_k) \in D_{Ts}} p_\theta(Y = y_k|\mathbf{x}_k)^{\text{occurrence number of}(\mathbf{x}_k, y_k)} \\
&= -\frac{1}{K} \log \prod_{(\mathbf{x}_k, y_k) \in D_{Ts}} p_\theta(Y = y_k|\mathbf{x}_k)^{ON(\mathbf{x}_k, y_k)} = -\frac{1}{K} \sum_{(\mathbf{x}_k, y_k) \in D_{Ts}} ON(\mathbf{x}_k, y_k) \log p_\theta(Y = y_k|\mathbf{x}_k) \\
&= -\sum_{(\mathbf{x}_k, y_k) \in D_{Ts}} \frac{ON(\mathbf{x}_k, y_k)}{K} \log p_\theta(Y = y_k|\mathbf{x}_k) = -\sum_{(\mathbf{x}_k, y_k) \in D_{Ts}} p_{D_{Ts}}(\mathbf{x}_k, y_k) \log p_\theta(Y = y_k|\mathbf{x}_k) \quad (4)
\end{aligned}
$$

where the last step comes from the fact that $(\mathbf{x}_k, y_k)$ is drawn from the testing distribution $D_{Ts}$. Continuing with Eq.(4), we obtain the first form of $\mathcal{L}(\theta, D_{Tr}, D_{Ts})$

$$
\begin{aligned}
\mathcal{L}(\theta, D_{Tr}, D_{Ts}) &= -\sum_k p_{D_{Ts}}(\mathbf{x}_k) p_{D_{Ts}}(Y = y_k|\mathbf{x}_k) \log \left( \frac{p_\theta(Y = y_k|\mathbf{x}_k)}{p_{D_{Ts}}(Y = y_k|\mathbf{x}_k)} \right) \\
&\quad - \sum_k p_{D_{Ts}}(\mathbf{x}_k) p_{D_{Ts}}(Y = y_k|\mathbf{x}_k) \log p_{D_{Ts}}(Y = y_k|\mathbf{x}_k) \\
&= \mathcal{E} \left[ p_{D_{Ts}}(y|\mathbf{x}) \log \frac{p_{D_{Ts}}(y|\mathbf{x})}{p_\theta(y|\mathbf{x})} \right] + \mathcal{H}(p_{D_{Ts}}(y|\mathbf{x})) \\
&= KL(p_{D_{Ts}}(y|\mathbf{x})||p_\theta(y|\mathbf{x})) + \mathcal{H}(p_{D_{Ts}}(y|\mathbf{x}))
\end{aligned}
$$

in which $KL(p_{D_{Ts}}(y|\mathbf{x})||p_\theta(y|\mathbf{x}))$ is the Kullback–Leibler divergence of $p_{D_{Ts}}(y|\mathbf{x})$ and $p_\theta(y|\mathbf{x})$, and $\mathcal{H}(p_{D_{Ts}}(y|\mathbf{x}))$ is the entropy of conditional distribution $p_{D_{Ts}}(y|\mathbf{x})$.

To obtain the second form of the log likelihood, we apply the chain rule of KL divergence

$$
KL(p(x, y)||q(x, y)) = KL(p(x)||q(x)) + KL(p(y|x)||q(y|x))
$$

And thus, we have

$$
\mathcal{L}(\theta, D_{Tr}, D_{Ts}) = KL(p_{D_{Ts}}(\mathbf{x}, y)||p_\theta(\mathbf{x}, y)) - KL(p_{D_{Ts}}(\mathbf{x})||p_\theta(\mathbf{x})) + \mathcal{H}(p_{D_{Ts}}(y|\mathbf{x}))
$$

## D. Proof of Theorem 7

Before proving the lemma, we note the following Theorems

**Theorem 12.** *(Pinsker's Inequality) ([Cover and Thomas, 2006])[Lemma 11.6.1] Given two probability mass functions p and q, the information/KL divergence is related to the variation distance via the following inequality*

$$KL(p||q) \geq \frac{1}{2ln2}||p-q||_1^2 = \frac{1}{2ln2}\left(\sum_{\mathbf{x},y}|p(\mathbf{x},y)-q(\mathbf{x},y)|\right)^2$$

**Theorem 13.** *(Convexity of Relative Entropy) ([Cover and Thomas, 2006])[Theorem 2.7.2] $KL(p||q)$ is convex in the pair $(p,q)$; that is, if $(p_1,q_1)$ and $(p_2,q_2)$ are two pairs of probability mass functions, then*

$$KL(\lambda p_1 + (1-\lambda)p_2||\lambda q_1 + (1-\lambda)q_2) \leq \lambda KL(p_1||q_1) + (1-\lambda)KL(p_2||q_2)$$

*for all $0 \leq \lambda \leq 1$.*

### D.1. Training Distribution Shift Scenario

We first prove the lower bound. From Eq.(3), we have

$$p_{D_{Tr}}(\mathbf{x}) = \sum_y p_{D_{Tr}}(\mathbf{x},y) = \alpha \sum_y p_{D_{Ts}}(\mathbf{x},y) + \beta \sum_y p_{\mathcal{D}_{tr,2}}(\mathbf{x},y) = \alpha p_{D_{Ts}}(\mathbf{x}) + \beta p_{\mathcal{D}_{tr,2}}(\mathbf{x})$$

and thus

$$p_{D_{Tr}}(y|\mathbf{x}) = \frac{p_{D_{Tr}}(\mathbf{x},y)}{p_{D_{Tr}}(\mathbf{x})} = \frac{\alpha p_{D_{Ts}}(\mathbf{x},y) + \beta p_{\mathcal{D}_{tr,2}}(\mathbf{x},y)}{\alpha p_{D_{Ts}}(\mathbf{x}) + \beta p_{\mathcal{D}_{tr,2}}(\mathbf{x})}$$

As $\mathcal{H}\left(p_{D_{Ts}}(y|\mathbf{x})\right)$ is a constant with respect to $\beta$ and can be ignored, we apply Theorem 12 into Eq.(1) to get

$$\mathcal{L}(\theta, D_{Tr}, D_{Ts}) \sim KL(p_{D_{Ts}}(y|\mathbf{x})||p_\theta(y|\mathbf{x})) = KL(p_{D_{Ts}}(y|\mathbf{x})||p_{D_{Tr}}(y|\mathbf{x}))$$

$$\geq \frac{1}{2ln2}\left(\sum_{\mathbf{x},y}|p_{D_{Ts}}(y|\mathbf{x}) - p_{D_{Tr}}(y|\mathbf{x})|\right)^2$$

Now,

$$|p_{D_{Ts}}(y|\mathbf{x}) - p_{D_{Tr}}(y|\mathbf{x})| = \left|p_{D_{Ts}}(y|\mathbf{x}) - \frac{\alpha p_{D_{Ts}}(\mathbf{x},y) + \beta p_{\mathcal{D}_{tr,2}}(\mathbf{x},y)}{\alpha p_{D_{Ts}}(\mathbf{x}) + \beta p_{\mathcal{D}_{tr,2}}(\mathbf{x})}\right|$$

$$= \left|\frac{\beta}{\alpha p_{D_{Ts}}(\mathbf{x}) + \beta p_{\mathcal{D}_{tr,2}}(\mathbf{x})}\right| \times |p_{D_{Ts}}(y|\mathbf{x})p_{\mathcal{D}_{tr,2}}(\mathbf{x}) - p_{\mathcal{D}_{tr,2}}(\mathbf{x},y)|$$

$$= \left|\frac{1}{\frac{\alpha}{\beta}p_{D_{Ts}}(\mathbf{x}) + p_{\mathcal{D}_{tr,2}}(\mathbf{x})}\right| \times |p_{D_{Ts}}(y|\mathbf{x})p_{\mathcal{D}_{tr,2}}(\mathbf{x}) - p_{\mathcal{D}_{tr,2}}(\mathbf{x},y)|$$

As $|p_{D_{Ts}}(y|\mathbf{x})p_{\mathcal{D}_{tr,2}}(\mathbf{x}) - p_{\mathcal{D}_{tr,2}}(\mathbf{x},y)|$, $p_{D_{Ts}}(\mathbf{x})$, and $p_{\mathcal{D}_{tr,2}}(\mathbf{x})$ are all $\geq 0$, it is straightforward to see that $|p_{D_{Ts}}(y|\mathbf{x}) - p_{D_{Tr}}(y|\mathbf{x})|$ is a monotonic non-decreasing function on $\beta$. As a consequence, we conclude that the lower-bound of loss function $\mathcal{L}(\theta, D_{Tr}, D_{Ts})$ is also a monotonic non-decreasing function on $\beta$.

To prove the upper bound, from Eq.(3), applying Theorem 13, we have

$$\begin{aligned}KL(p_{D_{Ts}}(\mathbf{x},y)||p_\theta(\mathbf{x},y)) &= KL(p_{D_{Ts}}(\mathbf{x},y)||p_{D_{Tr}}(\mathbf{x},y)) \\ &\leq \alpha KL(p_{D_{Ts}}(\mathbf{x},y)||p_{D_{Ts}}(\mathbf{x},y)) + \beta KL(p_{D_{Ts}}(\mathbf{x},y)||p_{\mathcal{D}_{tr,2}}(\mathbf{x},y)) \\ &= \beta KL(p_{D_{Ts}}(\mathbf{x},y)||p_{\mathcal{D}_{tr,2}}(\mathbf{x},y))\end{aligned}$$

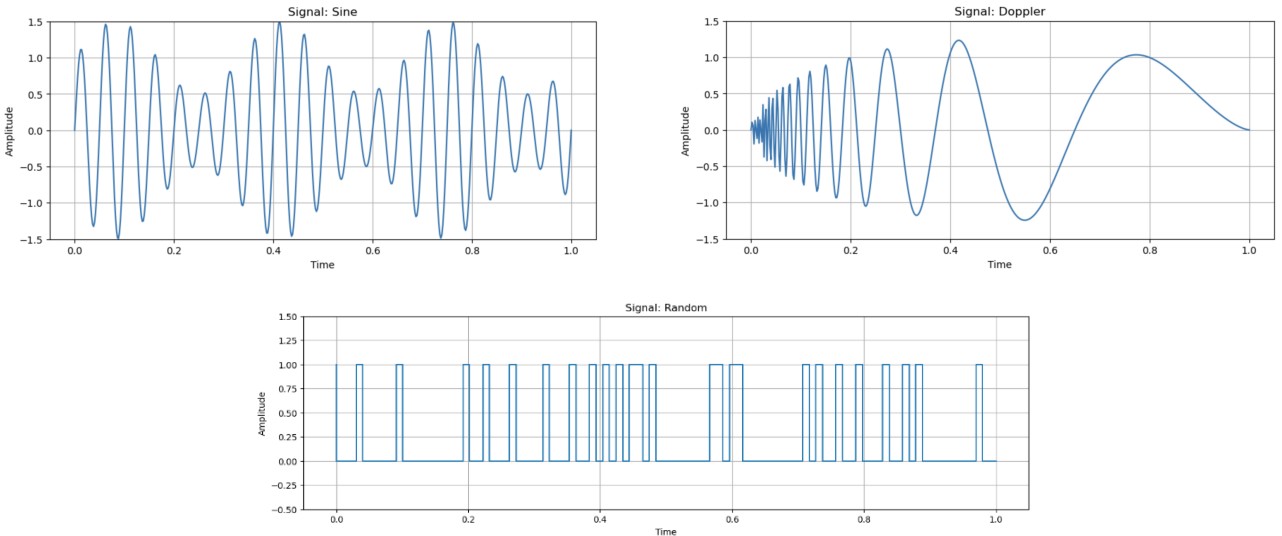

Figure 4: *Groundtruth signals used for experiments.*

Applying Theorem 12 to the KL divergence of the feature distribution, we have

$$KL(p_{D_{Ts}}(\mathbf{x})||p_\theta(\mathbf{x})) = KL(p_{D_{Ts}}(\mathbf{x})||p_{D_{Tr}}(\mathbf{x}))$$

$$\geq \frac{1}{2ln2}\left(\sum_{\mathbf{x}}|p_{D_{Ts}}(\mathbf{x})-p_{D_{Tr}}(\mathbf{x})|\right)^2$$

$$= \frac{\beta}{2ln2}\left(\sum_{\mathbf{x}}|p_{D_{Ts}}(\mathbf{x})-p_{\mathcal{D}_{tr,2}}(\mathbf{x})|\right)^2$$

Now as $\mathcal{H}\left(p_{D_{Ts}}(y|\mathbf{x})\right)$ is a constant with respect to $\beta$, we consider Eq.(2) to get

$$\mathcal{L}(\theta, D_{Tr}, D_{Ts})$$
$$\sim KL(p_{D_{Ts}}(\mathbf{x},y)||p_\theta(\mathbf{x},y)) - KL(p_{D_{Ts}}(\mathbf{x})||p_\theta(\mathbf{x}))$$

$$\leq \beta KL(p_{D_{Ts}}(\mathbf{x},y)||p_{\mathcal{D}_{tr,2}}(\mathbf{x},y)) - \frac{\beta}{2ln2}\left(\sum_{\mathbf{x}}|p_{D_{Ts}}(\mathbf{x})-p_{\mathcal{D}_{tr,2}}(\mathbf{x})|\right)^2$$

$$= \beta\left[KL(p_{D_{Ts}}(\mathbf{x},y)||p_{\mathcal{D}_{tr,2}}(\mathbf{x},y)) - KL(p_{D_{Ts}}(\mathbf{x})||p_{\mathcal{D}_{tr,2}}(\mathbf{x}))\right]$$

$$+ \beta\left[KL(p_{D_{Ts}}(\mathbf{x})||p_{\mathcal{D}_{tr,2}}(\mathbf{x})) - \frac{1}{2ln2}\left(\sum_{\mathbf{x}}|p_{D_{Ts}}(\mathbf{x})-p_{\mathcal{D}_{tr,2}}(\mathbf{x})|\right)^2\right]$$

Note that both terms on the right hand side are $\geq 0$ due to the chain rule of KL divergence and Pinsker's inequality. Therefore, the upper-bound of loss function $\mathcal{L}(\theta, D_{Tr}, D_{Ts})$ is a monotonic non-decreasing function of $\beta$.

### D.2. Testing Distribution Shift Scenario

The proof for this case is similar to that of Shifted Training Data Scenario, by swapping $p_{D_{Ts}}$ and $p_{D_{Tr}}$.

## E. Further Empirical Results

In Lemma 1 a bound that connects the estimation error to the degree of sparsity of the wavelet coefficients was established. The bound is general enough to support any chosen wavelet basis. When we choose the wavelet system to be Haar, the bound

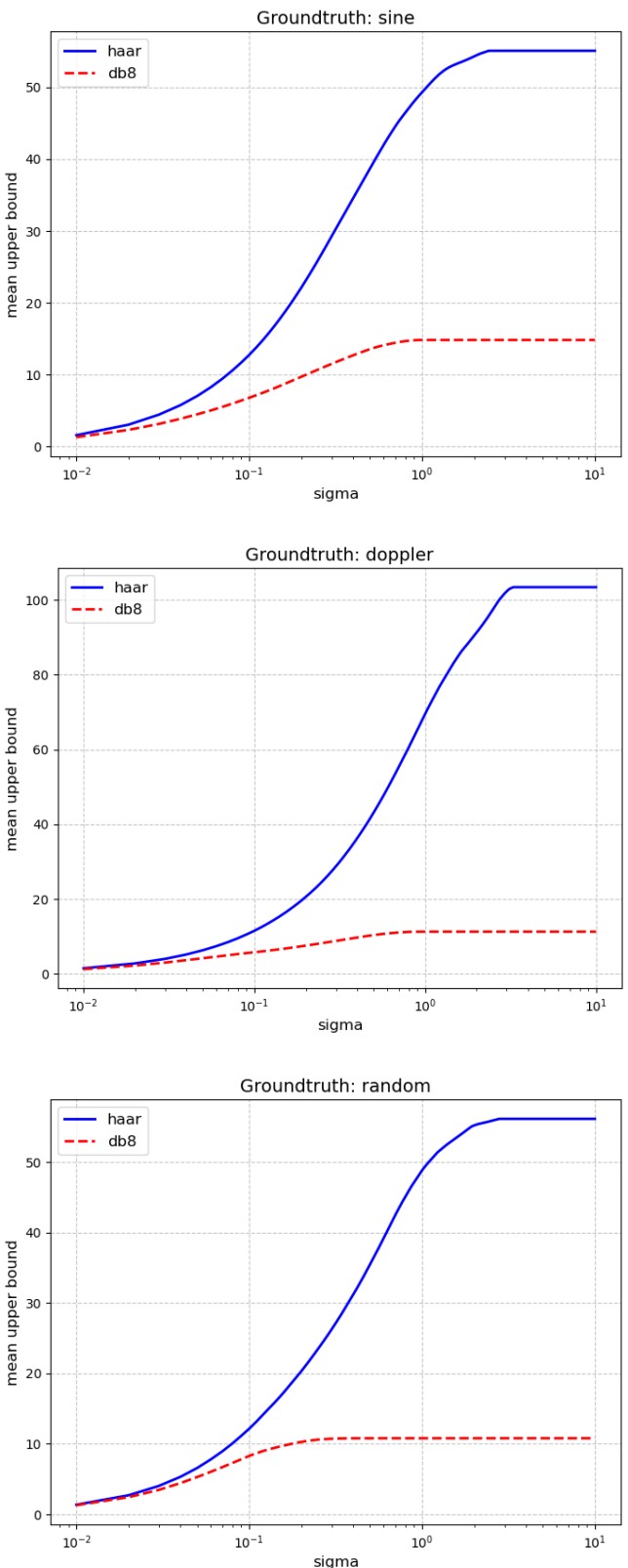

Figure 5: *Bound in Lemma 1 averaged across all time-stamps vs different noise levels in a semi-log plot. The groundtruth signals that are used are displayed in Fig.4. We can see that using a higher order wavelet like DB8 results in significantly lower values for the bound. This provides empirical evidence for the phenomenon where Lemma 1 can lead to sharper rates than that obtained by Theorem 2. We remind the reader that the bound in Theorem 2 is only an upperbound of the bound in Lemma 1 applied to Haar wavelets.*

in Lemma 1 can be further upperbounded by the variational expression displayed in Theorem 2. However it was argued in Section 2.2 that by choosing other wavelet systems, the bound in Lemma 1 can potentially lead to sharper error rates than that can be obtained by the Haar system. In this section, we give empirical evidence for this claim. The experimental setup is exactly same as described in Section 6.1. We report the value of the bound in Lemma 1 averaged across all timestamps vs different noise standard deviations in Figure 5. Experimental results similar to that of Section 6.1 is reported in Tables 4 and 5.

| Noise Level | AVG | HAAR (*This paper*) | DB8 (*This paper*) |
|---|---|---|---|
| 0.2 | $0.539 \pm 0.0011$ | $0.110 \pm 0.0019$ | $\mathbf{0.019 \pm 0.0005}$ |
| 0.3 | $0.541 \pm 0.0017$ | $0.170 \pm 0.0044$ | $\mathbf{0.031 \pm 0.0008}$ |
| 0.5 | $0.542 \pm 0.0027$ | $0.249 \pm 0.0070$ | $\mathbf{0.078 \pm 0.0018}$ |
| 0.7 | $0.544 \pm 0.0037$ | $0.247 \pm 0.0101$ | $\mathbf{0.141 \pm 0.0039}$ |
| 1 | $0.547 \pm 0.0052$ | $0.238 \pm 0.0154$ | $\mathbf{0.264 \pm 0.0065}$ |

Ground truth: Sine

Table 4: MSE of different algorithms when the noise standard deviation is known.

| Noise Level | AVG | ARW | Aligator | HAAR (*This paper*) | DB8 (*This paper*) |
|---|---|---|---|---|---|
| 0.2 | $0.538 \pm 0.0023$ | $0.542 \pm 0.0009$ | $0.435 \pm 0.0037$ | $0.152 \pm 0.0046$ | $\mathbf{0.029 \pm 0.0017}$ |
| 0.3 | $0.538 \pm 0.0035$ | $0.541 \pm 0.0014$ | $0432 \pm 0.0056$ | $0.184 \pm 0.0090$ | $\mathbf{0.037 \pm 0.0029}$ |
| 0.5 | $0.538 \pm 0.0057$ | $0.538 \pm 0.0021$ | $0.427 \pm 0.0093$ | $0.245 \pm 0.0140$ | $\mathbf{0.058 \pm 0.0061}$ |
| 0.7 | $0.539 \pm 0.0078$ | $0.526 \pm 0.0035$ | $0.422 \pm 0.0129$ | $0.265 \pm 0.0150$ | $\mathbf{0.085 \pm 0.0098}$ |
| 1 | $0.541 \pm 0.0108$ | $0.508 \pm 0.0048$ | $0.415 \pm 0.0182$ | $0.272 \pm 0.0225$ | $\mathbf{0.146 \pm 0.0163}$ |

Ground truth: Sine

Table 5: MSE of different algorithms when the noise standard deviation is unknown.

## F. Other Omitted Proofs

**Lemma 1.** *Consider the observation model $y_i = \theta_i + \epsilon_i$, for $i = 1, \ldots, n$ with $\epsilon_i$ being iid $\sigma$-sub-gaussian random variables. Let $\boldsymbol{y} := [y_n, \ldots, y_1]^T$, $\boldsymbol{\theta} = [\theta_n, \ldots, \theta_1]$ and $\boldsymbol{W}$ be an orthonormal wavelet transform matrix. Let $\tilde{\boldsymbol{\beta}} := \boldsymbol{W}\boldsymbol{y}$ and $\boldsymbol{\beta} := \boldsymbol{W}\boldsymbol{\theta}$ be respectively the empirical and true wavelet coefficients. Let $\mathcal{I}$ be an index set of wavelet coefficients that affect the value of reconstruction of the last groundtruth $\theta_1$. Let $W_{i,n}$ be the value of the element in $i^{th}$ row and $n^{th}$ column of $\boldsymbol{W}$. Let $\hat{\theta}_1$ be the estimate of the groundtruth $\theta_1$ obtained via Algorithm 1 with $\lambda = 2\sigma\sqrt{2\log(\log n/\delta)}$. Define $(a \wedge b) := \min\{a, b\}$, we have with probability at-least $1 - \delta$ that*

$$|\hat{\theta}_1 - \theta_1| \leq \sum_{i \in \mathcal{I}} 6|W_{i,n}| \cdot (|\beta_i| \wedge \lambda).$$

*Proof.* We start by noticing a bound on the maximum of $\log_2 n$ sub-gaussian random variables. Let $\epsilon_1, \ldots, \epsilon_{\log_2 n}$ be iid $\sigma$-sub-gaussian random variables. By sub-gaussian tail inequality we have for a fixed $i$,

$$P(|\epsilon_i| > t) \leq 2\exp(-t^2/(2\sigma^2)).$$

Now taking a union bound across all $\log_2 n$ random variables yields

$$P(\max_i |\epsilon_i| > t) \leq 2\log_2 n \exp(-t^2/2\sigma^2).$$

Hence we conclude that with probability at-least $1 - \delta$, we have

$$\max_i |\epsilon_i| \leq \sigma\sqrt{2\log(\log n/\delta)}. \tag{5}$$

We have $\tilde{\boldsymbol{\beta}} := \boldsymbol{W}\boldsymbol{y} = \boldsymbol{W}\boldsymbol{\theta} + \tilde{\boldsymbol{\epsilon}}$, where $\tilde{\boldsymbol{\epsilon}}$ are again drawn from a spherical Normal distribution. Let $\boldsymbol{\beta} = \boldsymbol{W}\boldsymbol{\theta}$ be the true wavelet coefficients.

Note that due to the dyadic MRA structure of the wavelet basis, only $\log_2 n$ wavelet coefficients are going to affect the value of the estimate $\hat{\theta}_1$. So $\mathcal{I}$ in the lemma statement becomes equal to a set $\{j_1, \ldots, j_{\log_2 n}\}$ for appropriatetly chosen indices. Let $\tilde{\epsilon}_{j_1}, \ldots, \tilde{\epsilon}_{j_{\log_2 n}}$ be the noise associated with those wavelet coefficients.

Let $\hat{\beta}_i = T_\lambda(\tilde{\beta}_i)$ with $\lambda = 2\sigma\sqrt{2\log(\log n/\delta)}$. Throughout the proof we use $\mathbb{I}$ to denote the binary indicator function.

For any $i \in \{j_1, \ldots, j_{\log_2 n}\}$, due to Eq.(5), we have with probability at-least $1 - \delta$ that $|\tilde{\epsilon}_i| \leq \lambda$. Now we proceed to bound the estimation error of soft-threshold in a coordinate-wise manner. We proceed in three cases:

Case 1: when $|\beta_i| \leq \lambda/2$. Then with probability at-least $1 - \delta$, we have that $|\tilde{\beta}_i| \leq \lambda$. So we must have $\hat{\beta}_i = 0$ with high probability. Consequently, $|\hat{\beta}_i - \beta_i| = |\beta_i| = \min\{|\beta_i|, \lambda/2\}$.

Case 2: when $|\beta_i| \geq 3\lambda/2$. Then with probability at-least $1 - \delta$ we must have $|\hat{\beta}_i - \beta_i| = |\tilde{\epsilon}_i - \lambda| \leq 2\lambda = 2\min\{|\beta_i|, \lambda\}$.

Case 3: when $\lambda/2 \leq |\beta_i| \leq 3\lambda/2$. Here we have,

Note that if $\text{sign}(\tilde{\beta}_i) = 1$ then $\text{sign}(\tilde{\beta}_i)(|\tilde{\beta}_i| - \lambda) - \beta_i = \tilde{\epsilon}_i - \lambda$. On the other hand if $\text{sign}(\tilde{\beta}_i) = -1$, then $\text{sign}(\tilde{\beta}_i)(|\tilde{\beta}_i| - \lambda) - \beta_i = \tilde{\epsilon}_i + \lambda$. Hence in both situations, we have with probability at-least $1 - \delta$ that $|\text{sign}(\tilde{\beta}_i)(|\tilde{\beta}_i| - \lambda) - \beta_i| \leq 2\lambda$. With this in mind, we proceed to bound the estimation error for Case 3.

$$
\begin{aligned}
|\hat{\beta}_i - \beta_i| &= |\beta_i|\mathbb{I}\{|\tilde{\beta}_i| \leq \lambda\} + (\text{sign}(\tilde{\beta}_i)(|\tilde{\beta}_i| - \lambda) - \beta_i)\mathbb{I}\{|\tilde{\beta}_i| > \lambda\} \\
&\leq 2\max\{|\beta_i|, \lambda\} \\
&\leq_{(a)} 3\lambda \\
&\leq 6\min\{|\beta_i|, \lambda\}
\end{aligned}
$$

where in the last line we used the fact that $\lambda/2 \leq |\beta_i|$ and in line (a) we used $|\beta_i| \leq 3\lambda/2$.

Combining all three cases leads to the conclusion that with probability at-least $1 - \delta$, we have for all $i \in j_1, \ldots, j_{\log_2 n}$,

$$
|\hat{\beta}_i - \beta_i| \leq 6\min\{|\beta_i|, \lambda\}.
$$

Now the statement of the lemma follows by reconstructing the last value of the signal from denoised wavelet coefficients and applying triangle inequality.

$\square$

**Lemma 14.** *Let the orthonormal Haar wavelet transform matrix be denoted by $\boldsymbol{H}$. Define quantities imilar as in lemma 1. We follow a dyadic indexing scheme to index elements in $\boldsymbol{\beta}$, i.e. $\boldsymbol{\beta} := [\beta_{-1}, \beta_{0,0}, \beta_{1,0}, \beta_{1,1}, \ldots, \beta_{j,0}, \beta_{j,1}, \ldots, \beta_{j,2^j-1}, \ldots, \beta_{\log_2 n-1,0}, \ldots, \beta_{\log_2 n-1,n/2-1}]^T \in \mathbb{R}^n$. Then the estimate $\hat{\theta}_1$ returned by algorithm 1 satisfies*

$$
|\hat{\theta}_1 - E[f(Z_1)]| \leq 3(\beta_{-1} \wedge \lambda)/\sqrt{n}
$$
$$
+ 3\sum_{i=0}^{\log_2 n-1} \frac{(|\beta_{i,2^i-1}| \wedge \lambda)}{\sqrt{n/2^i}},
$$

*with probability at-least $1 - \delta$.*

*Proof.* Consider the last column $\boldsymbol{h}_n$, of the Haar wavelet transform matrix. We can also index the last column using a dyadic indexing scheme as:

$\boldsymbol{h}_n = [h_{-1}, h_{0,0}, h_{1,0}, h_{1,1}, \ldots, h_{j,0}, h_{j,1}, \ldots, h_{j,2^j-1}, \ldots, h_{\log_2 n-1,0}, \ldots, h_{\log_2 n-1,n/2-1}]^T \in \mathbb{R}^n$. From the properties of the Haar matrix, we have $h_{-1} = 1/\sqrt{n}$, $|h_{j,k}| = \mathbb{I}\{k = 2^j - 1\}/\sqrt{n/2^j}$ where $\mathbb{I}$ is the binary indicator function.

With this structure in place, the proof the lemma is a direct consequence of Lemma 1.

$\square$

**Theorem 2.** *Let* $\bar{\theta}_{t:1} = (\theta_1 + \ldots + \theta_t)/t$. *For a time-point* $r$, *let* $S(r) := \{1, 2, 4, \ldots, 2^{\lfloor \log_2 r \rfloor}\}$. *Let* $U(r) := \max_{t \in S(r)} |\bar{\theta}_{t:1} - \theta_1|) \vee \sigma/\sqrt{r}$ *and* $\min_{r \in \{1, \ldots, n\}} U(r) := U(r^*)$ *with* $r^*$ *being the* smallest *time-point where the equality holds. By using the Haar wavelet system in Algorithm 1, we have with probability at-least* $1 - \delta$ *that*

$$|\hat{\theta}_1 - \theta_1| \leq \kappa \cdot U(r^*),$$

*where* $\kappa = (4\sqrt{2\log(\log n/\delta)} \vee 2\sqrt{2})(\log_2 n + 1)$ *and* $(a \vee b) := \max\{a, b\}$.

*Proof.* Suppose that maximum value among the *ordered set* $\mathcal{S} := \{(|\beta_{-1}| \wedge \lambda)/\sqrt{n}, (|\beta_{0,0}| \wedge \lambda)/\sqrt{n}, (|\beta_{1,1}| \wedge \lambda)/\sqrt{n/2}, \ldots,$
$(|\beta_{\log_2 n - 1, n/2 - 1}| \wedge \lambda)/\sqrt{2}\}$ occurs at the index $i^*$

With any index in the ordered set, we can associate a value of the time-scale which is basically the starting time-scale. For example, we associate the time-point index 1 with time-scale $n$, index 2 with time-scale $n$, index 3 with time-scale $n/2$, and so on untill index $\log_2 n + 1$ with time-scale 2. Let $t^*$ be the time-scale associated with the index $i^*$.

When the minimum in the set $\mathcal{S}$ is attained at $(|\beta_{-1}| \wedge \lambda)/\sqrt{n}$, then due to Lemma 14, the error $|\hat{\theta}_1 - E[f(Z_1)]| \leq 2\log_2 n\lambda/\sqrt{n} \leq 4\log_2 n\sqrt{2\log(\log n/\delta)}U(r)$ for any $r$ and the desired conclusion is trivial. So in the rest of the proof, we focus on the case where the minimum in the set $\mathcal{S}$ is *not* attained at $(|\beta_{-1}| \wedge \lambda)/\sqrt{n}$.

Let $j^* \leq r^* \leq 2j^*$ for some $j^*$ which is a power of 2. We consider two cases:

**Case (a):** when $t^* \geq j^*$.

We have that

$$\begin{aligned}
U(r^*) := (\max_{t \in S(r^*)} |\bar{\theta}_{t:1} - \theta_1|) \vee \sigma/\sqrt{r^*}) \\
\geq \sigma/\sqrt{r^*} \\
\geq \sigma/\sqrt{2j^*}.
\end{aligned} \tag{6}$$

Let the maximum value in the set $\mathcal{S}$ be $(|\beta| \wedge \lambda)/\sqrt{t^*}$ where $\beta$ is the value of the wavelet coefficient at the index $i^*$ in the set $\mathcal{S}$ and $t^*$ is its associated time-scale. Observe that

$$\begin{aligned}
\frac{(|\beta| \wedge \lambda)}{\sqrt{t^*}} &\leq 2\sigma\sqrt{\frac{2\log(\log n/\delta)}{t^*}} \\
&\leq_{(a)} 2\sigma\sqrt{\frac{2\log(\log n/\delta)}{j^*}} \\
&= 4\sigma\sqrt{\frac{\log(\log n/\delta)}{2j^*}} \\
&\leq 4\sqrt{\log(\log n/\delta)}U(r^*),
\end{aligned}$$

where in line (a) we used the fact that $t^* \geq j^*$ for the case considered and in the last line we used Eq.(6)

Now we can use Lemma 14 to upper bound the estimation error of algorithm 1 as

$$\begin{aligned}
|\hat{\theta}_1 - E[f(Z_1)]| &\leq (\log_2 n + 1)\frac{(\beta \wedge \lambda)}{\sqrt{t^*}} \\
&\leq 4(\log_2 n + 1)\sqrt{\log(\log n/\delta)}U(r^*),
\end{aligned}$$

**Case (b):** when $j^* > t^*$ We split the analysis into two scenarios:

**Scenario 1:** $U(r^*) = \sigma/\sqrt{r^*}$ The immediate conclusion for this scenario is:

$$\max_{t \leq r^*} |\bar{\theta}_{t:1} - \theta_1| \leq \frac{\sigma}{\sqrt{r^*}} \tag{7}$$

**Scenario 2:** $U(r^*) = \max_{t \leq r^*} |\bar{\theta}_{t:1} - \theta_1|$

In this scenario, we immediately obtain

$$U(r^*) \geq \sigma/\sqrt{r^*}$$

Due to the optimality of $U(r^*)$, we have

$$\max_{t \leq r^*} |\bar{\theta}_{t:1} - \theta_1| < \max_{t \leq r^*-1} |\bar{\theta}_{t:1} - \theta_1| \vee \frac{\sigma}{\sqrt{r^*-1}},$$

Recall that $r^*$ is the smallest optimal time-point (see the statement of the Theorem). Observe that $\max_{t \leq r^*-1} |\bar{\theta}_{t:1} - \theta_1| \vee \sigma/\sqrt{r^*-1}$ must be equal to $\sigma/\sqrt{r^*-1}$. Otherwise $r^*$ will not be the smallest optimal time-point. So in this scenario, we obtain

$$\max_{t \leq r^*} |\bar{\theta}_{t:1} - \theta_1| < \frac{\sigma}{\sqrt{r^*-1}} \tag{8}$$

Hence for case(b), we can conclude from Eq.(7) and (8) that

$$\max_{t \leq r^*} |\bar{\theta}_{t:1} - \theta_1| < \frac{\sigma}{\sqrt{r^*-1}}, \tag{9}$$

and

$$U(r^*) \geq \sigma/\sqrt{r^*}. \tag{10}$$

Let $\beta$ be the Haar wavelet coefficient corresponding to the index $i^*$ (with $t^*$ being the time-scale associated with that index as defined earlier) in the ordered set $\mathcal{S}$. We have

$$\begin{aligned}
\frac{|\beta|}{\sqrt{t^*}} &= |\bar{\theta}_{t^*:1} - \bar{\theta}_{t^*/2:1}| \\
&\leq |\bar{\theta}_{t^*:1} - \theta_1| + |\bar{\theta}_{t^*/2:1} - \theta_1| \\
&\leq 2 \max_{t \leq t^*} |\bar{\theta}_{t:1} - \theta_1| \\
&\leq_{(a)} 2 \max_{t \leq r^*} |\bar{\theta}_{t:1} - \theta_1| \\
&\leq \frac{2\sigma}{\sqrt{r^*-1}},
\end{aligned}$$

where in the last line we used Eq,(9) and in line (a) we used the fact that $t^* < j^* \leq r^*$ in Case (b) under consideration.

Hence we can conclude that

$$\begin{aligned}
\frac{(|\beta| \wedge \lambda)}{\sqrt{t^*}} &\leq \frac{|\beta|}{\sqrt{t^*}} \\
&\leq \frac{2\sigma}{\sqrt{r^*-1}} \\
&\leq \frac{2\sqrt{2}\sigma}{\sqrt{r^*}} \\
&\leq 2\sqrt{2}U(r^*),
\end{aligned}$$

where in the last line we used Eq.(10).

Now we can use Lemma 14 to upper bound the estimation error of algorithm 1 as

$$|\hat{\theta}_1 - E[f(Z_1)]| \leq (\log_2 n + 1)\frac{(\beta \wedge \lambda)}{\sqrt{t^*}}$$
$$\leq 2\sqrt{2}(\log_2 n + 1)U(r^*),$$

$\square$

**Corollary 3.** *For a sequence $\theta_{a:b}$ ($a > b$), define $TV(\theta_{a:b}) := \sum_{j=b+1}^{a}|\theta_j - \theta_{j-1}|$. For a time-point $r$, let $S(r) := \{1, 2, 4, \ldots, 2^{\lfloor \log_2 r \rfloor}\}$. Let $\tilde{U}(r) := \max_{t \in S(r)}|TV(\theta_{t:1}) - \theta_1|) \vee \sigma/\sqrt{r}$ and $\min_{r \in \{1,\ldots,n\}} \tilde{U}(r) := \tilde{U}(r^*)$ with $r^*$ being the smallest time-point where the equality holds. Define $\kappa = (4\sqrt{2\log(\log n/\delta)} \vee 2\sqrt{2})(\log_2 n + 1)$. By using CDJV wavelet transform with 2 vanishing moments in algorithm 1, we have with probability at-least $1 - \delta$ that*

$$|\hat{\theta}_1 - \theta_1| = O\left(\min\left\{\sum_{i \in \mathcal{I}}6|W_{i,n}| \cdot (|\beta_i| \wedge \lambda), \kappa\tilde{U}(r^*)\right\}\right).$$

*Proof.* The bound $\sum_{i \in \mathcal{I}}6|W_{i,n}| \cdot (|\beta_i| \wedge \lambda)$ is immediate from Lemma 1.

The proof of the Corollary follows mainly the steps of the proof of Theorem 2 with $U(\cdot)$ replaced by $\tilde{U}(\cdot)$. We elaborate on the arguments that take deviation from the former proof.

The departure in the proof steps happens in the way we bound $|\beta|/\sqrt{t^*}$ in case (b) of the proof of Theorem 2. We follow the follow the dyadic indexing scheme and notations defined the previous proof.

We focus on the case where $i^* > 2$. Otherwise we can bound the error optimally using similar logic we used to bound the error when $i^* = 1$ in the proof of Theorem 2.

$$\frac{|\beta|}{\sqrt{t^*}} \leq \sup_{j \geq 1}\sqrt{\frac{2^{j/2}}{n}}\|\beta_{j,0:2^j-1}\|_1$$
$$= O(TV(\theta_{t^*:1}))$$

where the last line is due to the fact that the class of functions of total variation less that some quantity $C$ is contained within a Besov space $B_{1,\infty}^1$ with radius $\nu C$ for some constant $\nu$ (Donoho et al., 1998; Johnstone, 2017).

The rest of the proof proceeds similarly as in the proof of Theorem 2.

$\square$

**Theorem 9.** *For a function $f \in \mathcal{F}$, we defined obtain its estimated loss $\hat{\ell}_f$ as follows. Run Algorithm 1 with data $\ell(f(\mathbf{x}_n), y_n), \ldots, \ell(f(\mathbf{x}_1), y_1)$ as input. Let $\hat{\ell}_f$ be the estimate returned by Algorithm 1. The ERM is defined by $\hat{f} \in argmin_{f \in \mathcal{F}}\hat{\ell}_f$.*

*For a function $f \in \mathcal{F}$, let $\boldsymbol{\theta}^f := [E[l(f(\mathbf{X}_n), y_n)], \ldots, E[l(f(\mathbf{X}_1), y_1)]]^T$. Let $\boldsymbol{\beta}^f := \mathbf{W}\boldsymbol{\theta}^f$ for a wavelet transform matrix $\mathbf{W}$. Consider the index set $\mathcal{I}$ defined in Lemma 1. Let $d$ be the VC dimension of $\mathcal{F}$. We have with probability at-least $1 - \delta$,*

$$L_{\hat{f}} - L_{f_*} \leq 8\sqrt{\frac{2d\log(2n)}{n}} + 2\sqrt{\frac{2\log(3/\delta)}{n}}$$
$$+ \sqrt{3} \cdot \sup_{f \in \mathcal{F}}\sum_{i \in \mathcal{I}}6|W_{i,n}| \cdot (|\beta_i^f| \wedge 2\sigma\sqrt{4d\log(3\log 2n/\delta)}).$$

*Proof.* We begin by recalling the definition of Shattering Coefficient (Mohri et al., 2012).

**Definition 15.** *Let $\mathcal{F}$ be a function that maps $(X, Y)$ to $0, 1$. The shattering coefficient is defined as the maximum number of behaviours over $n$ points.*

$$S(\mathcal{F}, n) := \max_{(x,y)_{1:n} \in \mathcal{X} \times \mathcal{Y}} |\{(\ell(f(x_n), y_n), \ldots, \ell(f(x_1), y_1)) : f \in \mathcal{F}\}|.$$

*We say that subset $\mathcal{F}' \subseteq \mathcal{F}$ is an $n$-shattering-set if it is a smallest subset of $\mathcal{F}$ such that for any $(\ell(f(x_n), y_n), \ldots, \ell(f(x_1), y_1))$ there exists some $f' \in \mathcal{F}'$ such that $(\ell(f(x_n), y_n), \ldots, \ell(f(x_1), y_1)) = (\ell(f'(x_n), y_n), \ldots, \ell(f'(x_1), y_1))$.*

By standard arguments, the excess risk can be bounded as

$$L_{\hat{f}} - L_{f_*} \leq |L_{\hat{f}} - \hat{\ell}_{\hat{f}}| + |L_{f_*} - \hat{\ell}_{f_*}| + \hat{\ell}_{\hat{f}} - \hat{\ell}_{f_*}$$
$$\leq |L_{\hat{f}} - \hat{\ell}_{\hat{f}}| + |L_{f_*} - \hat{\ell}_{f_*}|.$$

Next, we need to bound quantities of the form $|L_f - \hat{\ell}_f|$ for any function $f \in \mathcal{F}$ so as to facilitate uniform convergence arguments later. However, observe that this is different from usual arguments done in statistical learning theory (Bousquet et al., 2004) because the estimate $\hat{\ell}_f$ is not a sum of individual losses.

Let $(x_1' y_1'), \ldots, (x_n', y_n')$ be $n$ ghost samples from the same distribution $D_1$. Let $\ell_{f,n} := \frac{1}{n} \sum_{i=1}^n \ell(f(x_i'), y_i')$ be the empirical loss for distribution $D_1$. We have

$$\sup_{f \in \mathcal{F}} |L_f - \hat{\ell}_f| \leq \sup_{f \in \mathcal{F}} |L_f - \ell_{f,n}| + \sup_{f \in \mathcal{F}} |\ell_{f,n} - \hat{\ell}_f|$$

The first term above can be bounded using a standard argument based on Rademacher complexity and a further application of Massart's and Sauer's lemmas (Mohri et al., 2012). More precisely with probability at-least $1 - \delta/3$, we have

$$\sup_{f \in \mathcal{F}} |\ell_{f,n} - L_f| \leq 4\sqrt{\frac{2d \log(n)}{n}} + \sqrt{\frac{2 \log(3/\delta)}{n}}. \tag{11}$$

Notice that the second term only depends on the observed and the ghost data-sets. Let $\mathcal{F}'$ be an $(2n)$-shattering set of $\mathcal{F}$.

$$\sup_{f \in \mathcal{F}} |\ell_{f,n} - \hat{\ell}_f| = \sup_{f \in \mathcal{F}'} |\ell_{f,n} - \hat{\ell}_f|$$
$$\leq \sup_{f \in \mathcal{F}'} |\ell_{f,n} - L_f| + \sup_{f \in \mathcal{F}'} |\hat{\ell}_f - L_f|$$

Define $\lambda' L = 2\sqrt{2 \log(3nS(\mathcal{F}, 2n)/\delta)}$. For the last term, we have by Lemma 1 and a union bound across $\mathcal{F}'$, with probability at-least $1 - \delta/3$ that

$$\sup_{f \in \mathcal{F}'} |\hat{\ell}_f - L_f| \leq \sup_{f \in \mathcal{F}'} \sum_{i \in \mathcal{I}} 6|W_{i,n}| \cdot (|\beta_i^f| \wedge \lambda' \sqrt{3})$$
$$\leq \sqrt{3} \cdot \sup_{f \in \mathcal{F}'} \sum_{i \in \mathcal{I}} 6|W_{i,n}| \cdot (|\beta_i^f| \wedge \lambda')$$
$$\leq_{(a)} \sqrt{3} \cdot \sup_{f \in \mathcal{F}'} \sum_{i \in \mathcal{I}} 6|W_{i,n}| \cdot (|\beta_i^f| \wedge 2\sigma\sqrt{4d \log(3 \log 2n/\delta)})$$
$$= \sqrt{3} \cdot \sup_{f \in \mathcal{F}} \sum_{i \in \mathcal{I}} 6|W_{i,n}| \cdot (|\beta_i^f| \wedge 2\sigma\sqrt{4d \log(3 \log 2n/\delta)}), \tag{12}$$

, where line (a) is due to Sauer's lemma (Sauer, 1972).

Using standard Rademacher complexity arguments, we have with probability at-least $1 - \delta/3$

$$
\begin{aligned}
\sup_{f \in \mathcal{F}'} |\ell_{f,n} - L_f| &\leq 4\sqrt{\frac{2 \log S(\mathcal{F}, 2n)}{n}} + \sqrt{\frac{2 \log(3/\delta)}{n}} \\
&\leq 4\sqrt{\frac{2d \log(2n)}{n}} + \sqrt{\frac{2 \log(3/\delta)}{n}},
\end{aligned}
\tag{13}
$$

where the last line is due to Sauer's Lemma. Now union bounding across Equations (11), (12) and (13) yields that with probability at-least $1 - \delta$ we have

$$
\begin{aligned}
L_{\hat{f}} - L_{f_*} &\leq 8\sqrt{\frac{2d \log(2n)}{n}} + 2\sqrt{\frac{2 \log(3/\delta)}{n}} \\
&\quad + \sqrt{3} \cdot \sup_{f \in \mathcal{F}} \sum_{i \in \mathcal{I}} 6|W_{i,n}| \cdot (|\beta_i^f| \wedge 2\sigma\sqrt{4d \log(3 \log 2n/\delta)}).
\end{aligned}
$$

$\square$

**Theorem 11.** *Suppose that an algorithm satisfies a bound of the form given in Theorem 2 or Corollary 3. Consider running the algorithm iteratively to produce the estimates $\hat{\theta}_{1:n}$. Then with probability at-least $1 - \delta$ we have that*

$$
R_{sq}(\hat{\theta}_{1:n}, \theta_{1:n}) = \tilde{O}(n^{1/3} C^{2/3} \sigma^{4/3})
$$
$$
R_{abs}(\hat{\theta}_{1:n}, \theta_{1:n}) = \tilde{O}(n^{2/3} C^{1/3} \sigma^{2/3}).
$$

*Proof.* For two time points $a < b$, let $C_{a \to b} := \sum_{t=a+1}^{b} |\theta_t - \theta_{t-1}|$ be the TV of the groundtruth sequence within an interval $[a, b]$. We begin my partitioning the horizon into $M$ bins as $[1_s, 1_t], \ldots, [i_s, i_t], \ldots, [M_s, M_t]$ wherein $i_s = (i-1)_t + 1$. For the bin number $i$, let the quantity $n_i := i_t - i_s + 1$ denote its length. The bins in the partition are constructed such that $C_{i_s \to i_t} \leq \sigma/\sqrt{n_i}$ while $C_{i_s \to i_t+1} > \sigma/\sqrt{n_i + 1}$ for all bins (except possibly for the last bin). We first bound the number of bins in the partition. We focus on the non-trivial case where $M > 1$. We have

$$
\begin{aligned}
C &\geq \sum_{i=1}^{M-1} C_{i_s \to i_t+1} \\
&\geq \sum_{i=1}^{M-1} \sigma/\sqrt{n_i + 1} \\
&\geq \sum_{i=1}^{M-1} \sigma/\sqrt{2n_i} \\
&\geq_{(a)} \frac{\sigma(M-1)}{\sqrt{2n/(M-1)}} \\
&\geq \frac{\sigma M^{3/2}}{4\sqrt{n}},
\end{aligned}
$$

where in the last line we used $M > 1$ so that $M - 1 \geq M/2$ and in line (a) we used Jensen's inequality along with the convexity of the function $f(x) = 1/\sqrt{x}$. Rearranging the last display yields $M \leq 4^{2/3} n^{1/3} C^{2/3} \sigma^{-2/3}$.

Next, we proceed to bound the estimation error. Consider a bin $[i_s, i_t]$ and let $j \in [i_s, i_t]$. Notice that $|\bar{\theta}_{i_s:j} - \theta_j| \leq C_{i_s \to j} \leq C_{i_s \to i_t}$. With this observation the bound in Theorem 2 can be re-written into simpler form for any $j \in [i_s, i_t]$ as

$$
\begin{aligned}
|\hat{\theta}_j - \theta_j| &\leq \kappa \left( C_{i_s \to i_t} \vee \sigma/\sqrt{j - i_s + 1} \right) \\
&\leq \kappa \cdot \sigma/\sqrt{j - i_s + 1},
\end{aligned}
$$

where in the last line we used the property of the partition that $C_{i_s \to i_t} \leq 1/\sqrt{n_i}$.

Hence we can upperbound the risk within bin $i$ as

$$
\begin{aligned}
R_{\mathrm{sq}}(\hat{\theta}_{i_s:i_t}, \theta_{i_s:i_t}) &\leq \sum_{k=1}^{n_i} \kappa^2 \cdot \sigma^2/k \\
&\leq 2\kappa^2 \log n_i.
\end{aligned}
$$

Now summing the risk across all $M$ bins and using the fact that $M \leq 4^{2/3} n^{1/3} C^{2/3} \sigma^{-2/3}$ yields $R_{\mathrm{sq}}(\hat{\theta}_{1:n}, \theta_{1:n}) \leq 2^{7/3} \kappa^2 \log n \cdot n^{1/3} C^{2/3} \sigma^{4/3}$.

Proceeding similarly, we can bound the risk in absolute deviations. We have

$$
\begin{aligned}
R_{\mathrm{abs}}(\hat{\theta}_{1:n}, \theta_{1:n}) &= \sum_{i=1}^{M} R_{\mathrm{abs}}(\hat{\theta}_{i_s:i_t}, \theta_{i_s:i_t}) \\
&\leq \sum_{i=1}^{M} \sum_{k=1}^{n_i} \kappa \cdot \sigma/\sqrt{k} \\
&\leq \sum_{i=1}^{M} 2\kappa\sqrt{n_i} \\
&\leq_{(a)} 2\sigma\kappa\sqrt{Mn} \\
&\leq 2\kappa n^{2/3} C^{1/3} \sigma^{2/3},
\end{aligned}
$$

wherein line (a) we used Cauchy–Schwarz inequality. $\qquad\square$

