# OpenReview forum: "Adaptive Estimation and Learning under Temporal Distribution Shift"
_ICML.cc/2025/Conference — ICML 2025 poster_

### Official Review · Reviewer_8JTP · 2025-03-12

**Overall Recommendation:** 2

**Summary:**

This paper focuses on estimation and learning time series data in the presence of temporal distribution shifts. The authors propose a wavelet soft-thresholding estimator that optimally estimates the ground truth sequence under unknown shifts and provide theoretical error bounds for their method. Their approach generalizes existing research by linking the sequence’s non-stationarity to sparsity in the wavelet domain. The paper also applies this estimator to binary classification under distribution shifts and establishes its connection to total-variation denoising. The authors conduct experiments on synthetic data to validate their proposed method.

**Claims And Evidence:**

The authors made several claims in their paper. However, some claims are not well supported:
- The authors claimed that using higher-order wavelets can achieve better performance. However, their theoretical analysis and algorithm do not provide a principled method for choosing the optimal wavelet transform.
- The authors claimed that their algorithm achieve better computational efficiency for binary classification setting compared to the existing works. However, no empirical evidence is provided to support it.
- The authors claimed the superior performance of their proposed wavelet-denoising based algorithms in estimating the ground-truth, compared to prior works. However, they lacks real-world data validation for it.

**Essential References Not Discussed:**

N/A

**Experimental Designs Or Analyses:**

- The authors primarily validate their method on synthetic datasets (e.g., Random and Doppler signals) rather than real-world time series data such as financial market data, climate trends, or network traffic patterns. This limits the generalizability of their findings to practical applications where temporal distribution shifts may be more complex and unpredictable.
- The study focuses on classical estimation and statistical learning techniques, omitting comparisons with modern deep learning approaches that incorporate adaptive architectures for handling temporal distribution shifts (e.g., transformer models or recurrent neural networks)
- The authors leverage higher-order wavelets (i.e., Daubechies-8) but does not thoroughly explore how different wavelet families affect estimation performance. Ablation study need to be conducted to explore this issue. Moreover, there is no discussion on when Haar wavelets suffice versus when higher-order wavelets provide advantages, leaving an open question about optimal wavelet selection.
- The impact of hyperparameter selection requires further discussion. Specifically, the authors rely on a fixed soft-thresholding approach for wavelet denoising but do not investigate how adaptive tuning could enhance performance.
- Experiment with binary classification setting is highly recommended to demonstrate the utility of the proposed method.

**Methods And Evaluation Criteria:**

- The authors proposed wavelet-denoising algorithm (Algorithm 1) for time series data estimation problem. However, it is not clear how to apply this algorithm for binary classification problem as shown in Theorem 9. Given that Algorithm 1 only takes Y as input instead of both X and Y. More explanations are needed.
- The algorithm does not provide a principled method for choosing the optimal wavelet transform. While Haar wavelets work well in many cases, higher-order wavelets may be needed for complex trends, but their selection remains ad hoc.
- The algorithm uses fixed soft-thresholding for wavelet denoising, which may not always be optimal.
- The algorithm assumes that the ground truth sequence has sparse wavelet coefficients. However, it's hard to verify this assumption in practical application.
- The temporal shift setting the algorithm considers is limited. Specifically,it does not continuously update its model as new data arrives (online learning).

**Other Comments Or Suggestions:**

Please see the above sections.

**Other Strengths And Weaknesses:**

Please see the above sections.

**Questions For Authors:**

Please see the above sections.

**Relation To Broader Scientific Literature:**

This paper falls within the fields of domain adaptation and time-series estimation, tackling challenges related to temporal distribution shifts in machine learning. However, its technical contributions and advantages over existing literature remain unclear. Please refer to other sections for further details.

**Theoretical Claims:**

- The theoretical analysis for binary classification setting (section 3) provide no new insights compared to existing works in domain adaptation literature. In particular, the error is bounded by the distance of the joint data distribution between training and testing datasets is widely known.
- While the paper claims that higher-order wavelets (e.g., Daubechies wavelets) can improve estimation, it does not provide a theoretical comparison between different wavelet families.
- The analysis is limited to a specific type of temporal shift, as the authors only consider scenarios where the training and testing data are modeled as a linear combination of two base distributions.

---

> ### Author Rebuttal · Authors · 2025-03-31
>
> **Experiments on real dataset**
>
> Please see the response to Reviewer c9dT under the same comment title.
>
> **How to do optimal wavelet selection.**
>
> This is a well-known model selection problem in wavelet denoising, not a drawback unique to our work. We acknowledge this in Section 7. Practitioners typically analyze data trends—if they follow a piecewise polynomial of degree $k$, a wavelet of order $k+1$ is chosen, as it effectively models such structures. However, higher-order wavelets introduce numerical instabilities and variance, though the latter can be mitigated with more data. Selecting a wavelet basis is akin to choosing a kernel in Gaussian Processes—application-specific and guided by practical intuition. A more comprehensive exposition can be obtained from [5].
>
> **Theoretical justification for using higher order wavelets**
>
> Please see the comment to Reviewer 4jCD under the same comment title.
>
> **Computational efficiency in comparison to prior work (Mazzetto and Upfal , 2023)**
>
> Applying the algorithm from (Mazzetto and Upfal , 2023) requires $O(\log n)$ calls to an ERM oracle. While our method requires exactly one call as detailed in Section 4. This is the reason for computational efficiency.
>
> **How to use Algorithm 1 for binary classification?**
>
> Please refer to the statement of Theorem 9. We compute refined the loss estimate of a model at most recent timestamp. For this, loss sequence $\ell(f(x_n),y_n),\ldots,\ell(f(x_1),y_1)$ is given as input to Algorithm 1. ERM is performed using the returned loss estimate.
>
> **Algorithm uses fixed soft-thresholding for wavelet denoising, which may not always be optimal**
>
> The guarantee in Theorem 2 is obtained via the universal soft-thresholding based denoising. The bound is known to be minimax optimal in light of (Mazzetto and Upfal , 2023).
>
> **The algorithm assumes that the ground truth sequence has sparse wavelet coefficients but it's hard to verify this assumption**
>
> We would like to point out that such an assumption is not made. Instead the bounds naturally adapts to the sparsity of the wavelet coefficient and hence the degree of non-stationarity of the groundtruth without any prior knowledge.
>
> **The temporal shift setting the algorithm considers is limited and not applicable to online learning...**
>
> We clarify that our setup is offline: given all data $y_n, \dots, y_1$, the goal is to estimate the ground truth at $t=1$. For online settings, Algorithm 1 can be used iteratively to update estimates over time. The multi-resolution nature of wavelets enables efficient updates with $O(\log n)$ complexity per round. Unlike typical online methods with cumulative regret guarantees, our approach provides stronger per-round estimation error guarantees.
>
> **Study omits comparisons with transformers and RNN architectures**
>
> Our method operates in a data-scarce regime. For example, to estimate the groundtruth at timestamp 50 we only have just 50 data points. Advanced models like transformers and RNNs excel with abundant data. Hence, we compare against methods suited for low-data settings that are also known to have robust theoretical guarantees on estimation error.
>
> **Results on adaptive tuning schemes for thresholding**
>
> As per reviewer’s suggestion, we conducted experiments on other well studied thresholding schemes namely SUREShrink [1] and Energy based thresholding [3] heuristic. The MSE results are reported below for Doppler signal and different wavelet types. Name in brackets indicates the thresholding scheme. Similar observations hold true for Random ground truth signal.
>
> | Noise Level | Haar (sure) | Haar (energy) | Haar (soft) |
> |-------|------------|--------------|-------------|
> | 0.2   | 0.0376 ± 0.0023 | 0.0210 ± 0.0010 | 0.053 ± 0.0017 |
> | 0.3   | 0.0400 ± 0.0031 | 0.0265 ± 0.0015 | 0.056 ± 0.0018 |
> | 0.5   | 0.0490 ± 0.0044 | 0.0456 ± 0.0024 | 0.065 ± 0.0018 |
> | 0.7   | 0.0653 ± 0.0051 | 0.0726 ± 0.0032 | 0.072 ± 0.0031 |
> | 1.0   | 0.0938 ± 0.0055 | 0.1265 ± 0.0054 | 0.088 ± 0.0035 |
>
> | Noise Level | DB8 (sure) | DB8 (energy) | DB8 (soft) |
> |-------|------------|--------------|-------------|
> | 0.2   | 0.0190 ± 0.0013 | 0.0195 ± 0.0007 | 0.0204 ± 0.0007 |
> | 0.3   | 0.0257 ± 0.0017 | 0.0274 ± 0.0010 | 0.0265 ± 0.0010 |
> | 0.5   | 0.0464 ± 0.0029 | 0.0527 ± 0.0015 | 0.0444 ± 0.0016 |
> | 0.7   | 0.0760 ± 0.0054 | 0.0904 ± 0.0023 | 0.070 ± 0.0021  |
> | 1.0   | 0.1406 ± 0.0090 | 0.1697 ± 0.0035 | 0.129 ± 0.0058  |
>
> This experiment shows that some thresholding schemes may outperform universal soft thresholding empirically, but no single method is best overall. While SUREShrink is known to achieve minimax MSE optimality in non-parametric regression, its extension to point-wise bounds remains unclear. We will include these insights in the manuscript.
>
> **References**
>
> [1] https://www.jstor.org/stable/2291512
>
> [3] https://digital-library.theiet.org/doi/full/10.1049/iet-smt.2016.0168
>
> [5] https://www.sciencedirect.com/book/9780123743701/a-wavelet-tour-of-signal-processing

---

### Official Review · Reviewer_c9dT · 2025-03-13

**Overall Recommendation:** 3

**Summary:**

This paper investigates the problem of learning under temporal distribution shift, where the task is to estimate the ground truth related to the last observation under minimal stationarity assumptions. They prove new bounds on existing versions of a soft-thresholding algorithm for the problem, and translate these findings to general classification and learning problems under temporal distribution shift. The key contribution of the work is that their bounds depend on the sum of the wavelet coefficients, which allow for fast rates when the sparsity level is high.

**Claims And Evidence:**

The claims are supported by clear and convincing evidence. I would appreciate more insight into the proof techniques for each of the Lemmas, but the results themselves are convincing.

**Essential References Not Discussed:**

None that I am aware of.

**Experimental Designs Or Analyses:**

The experimental methodology seems sound.

**Methods And Evaluation Criteria:**

Synthetic datasets are provided, and the paper’s approach is shown to outperform the state of the art in a number of cases. The paper could benefit from a real data experiment or experimental application.

**Other Comments Or Suggestions:**

N/A

**Other Strengths And Weaknesses:**

I think the paper is mostly clear, with a few minor drawbacks related to clarity:

- Algorithm 1, what is H? not defined
- Thm 2 “syetm”

- Lemma 4: not clear whether this is an already known result, this appears very foundational, please cite if it came from somewhere else or state that this is a previously unknown result and an original contribution.
- Def 6: I do not understand the last line, could you explain that more clearly? WIth the T1=Ts if T = Tr, etc.
- Paragraph before section 4 is not clear to me, with a few spelling/grammar mistakes.
- Theorem 9 typo “defined obtain”
- First paragraph of section 5, “has been not uncovered” typo/unclear

**Questions For Authors:**

1) Below the definition of the total variation class, you define the alternate sequence penalising the sum of squared differences, but the example you give is not the sum of squared differences right? Is this a typo?

**Relation To Broader Scientific Literature:**

The paper considers an important and foundational problem in statistics and machine learning, and relates its findings to previous related works on the Total Variation Denoising problem tackled by Van de Geer and more immediately related works such as Mazzetto and Upfal 2023. The paper builds upon these previous works and expands the literature on the problem by proving new upper bounds on the risk of estimation under temporal distribution shift, and highlight the adaptive nature of such wavelet based methods.

The main contribution of the paper as per my understanding is the precise quantification of the role of sparsity in the accuracy of these wavelet-based methods. It does not, as far as I know, propose an entirely new methodology or algorithm, so its contribution is mainly a useful theoretical one to better understand existing algorithms in the field.

**Theoretical Claims:**

I did not check the correctness of proofs, and the lack of proof ideas/techniques makes it hard to verify the accuracy of proofs in the main body. The results themselves are convincing and not completely surprising, leading me to believe that the theoretical claims are not overstated.

---

> ### Author Rebuttal · Authors · 2025-03-31
>
> We thank the reviewer for their comments. Please see the responses below.
>
> **Experiments on real dataset**
>
> As an application of our proposed methods, we conduct a model selection experiment using real-world data.
> We evaluate our method on data from the Dubai Land Department (https://www.dubaipulse.gov.ae/data/dld-transactions/dld_transactions-open), following the setup identical to that of [2]. The dataset includes apartment sales from January 2008 to December 2023 (192 months). Each month is treated as a time period, where the goal is to predict final prices based on apartment features. Data is randomly split into 20\% test, with two train-validation splits: (a) 79\%-1\% and (b) 75\%-5\%.
>
> For each month $t$, we train Random Forest and XGBoost models using a window of past data where we consider window sizes $w \in [1,4,16,64,256]$, yielding 10 models per month. Validation MSEs from past and current months are used to refine the current month’s estimate of MSE via Algorithm 1 or ARW from [2]. The refined validation scores are used to select the best model for final MSE evaluation on test data. We report the average MSE of this model selection scheme over 192 months and 5 independent runs, comparing our method to ARW [2]. In the table below, HAAR and DB8 are versions of Algorithm 1 with the corresponding wavelet basis given as input.
>
> Case 1:  79\%-1\% Train-Validation split for each month
> --
> | Method  | MSE ± Std. Error |
> |---------|------------------|
> | **ARW (from [2])**  | $0.079 \pm 0.0005$  |
> | **HAAR (ours)** | $0.0722 \pm 0.0006$ |
> | **DB8 (ours)**  | $0.0762 \pm 0.0002$ |
>
> Case 2:  75\%-5\% Train-Validation split for each month
> --
> | Method  | MSE ± Std. Error |
> |---------|------------------|
> | **ARW (from [2])**  | $0.0719 \pm 0.0005$  |
> | **HAAR (ours)** | $0.0736 \pm 0.0011$ |
> | **DB8 (ours)**  | $0.0768 \pm 0.0008$ |
>
> We see that wavelet based methods shine especially when the validation data is scarce. This allows us to include more data for training while still allowing to obtain high quality estimates for validation scores. Such a property can be especially helpful in data-scarce regimes. Unlike the synthetic data experiments, here we find that Haar wavelets perform better than DB8. This can be attributed to the following facts: i) the noise in the observations depart from iid sub-gaussian assumption and; ii) the high degree of non-stationarity in the pricing data as indicated by Fig.6(b) in [2] makes the underlying trends to have a low degree piecewise polynomial structure. This makes the groundtruth irregular (or less smooth) which can be suitably handled by lower order Haar wavelets which are also less smooth and abrupt (see Fig. 2 in Appendix).
>
> **Lemma 4: please cite if it came from somewhere else or state that this is a previously unknown result**
>
> We have mentioned in the proof that Lemma 4 that it is adapted from (Achille and Soatto, 2018). For clarity, we will add the phrase “(adapted from Achille and Soatto, 2018)” in the Lemma statement itself.
>
> **Def 6: I do not understand the last line, could you explain that more clearly? With the T1=Ts if T = Tr, etc**
>
> Thanks for pointing this out! There is a typo and the sentence should be “Here T represents training data T≡T_r (or testing data T≡T_s) and T1 ≡ T_s (or T1 ≡ T_r).” We will revise the manuscript accordingly. We will also add the following explanation into the manuscript.
> “Specifically, under the Training Distribution Shift Scenario, the training data T≡T_r consists of (1) samples from the testing distribution T_1≡T_s, and (2) samples from a dissimilar distribution T_2. When, more and more dissimilarly-distributed samples from T_2 are added into the training distribution, this leads to higher dissimilarity ratio $\beta$.
> Under the Testing Distribution Shift Scenario, the testing data T≡T_s consists of (1) samples from the training distribution T_1≡T_r, and (2) samples from a dissimilar distribution T_2. When, more and more dissimilarly-distributed samples from T_2 are added into the testing distribution, this leads to higher dissimilarity ratio $\beta$.
>
> **Typos**
>
> Thanks for pointing out the typos. We will fix them. $H$ in Algorithm 1 should be $W$ the matrix for wavelet transform. The example below the definition of the TV class must be sum of squared differences as the reviewer pointed out.  This is a typo that will be corrected.
>
> **References**
>
> [1] Adapting to Unknown Smoothness via Wavelet Shrinkage, David L. Donoho and Iain M. Johnstone, Journal of the American Statistical Association, 1995
>
> [2] Model Assessment and Selection under Temporal Distribution Shift, Elise Han, Chengpiao Huang, and Kaizheng Wang, ICML 2024

---

> > ### Comment · Reviewer_c9dT · 2025-04-06
> >
> > Thank you for your responses. I found the real data experiment useful and a good addition to the paper. I will maintain my score.

---

### Official Review · Reviewer_4jCD · 2025-03-17

**Overall Recommendation:** 3

**Summary:**

The paper studies the problem of temporal distribution shift, by formulating the problem as parameter estimation in an univariate non-stationary sequence with sub-gaussian noise. The authors propose a wavelet denoising approach for this estimation problem, and theoretically upper bound its error rate. The authors then show how to use this algorithm as a subroutine for learning using ERM. Finally, the authors show that their method outperforms a sliding window baseline from prior work on synthetic data.


## update after rebuttal

I thank the authors for their rebuttal. Most of my concerns have been addressed, except for the limitations of the univariate setting. I will keep my score.

**Claims And Evidence:**

1. The authors empirically show that higher order wavelets (DB8) outperforms the Haar wavelets. However, there does not seem to be theoretical justification for why this happens. Having a toy example to demonstrate this would also be helpful.

2. The authors study a very simple system with a univariate time series. How would the theorems and algorithm be extended to the multivariate setting?

3. The authors hint (at the end of Section 4) that it possible to solve the ERM problem using a differentiable surrogate loss, but this is not expanded on further methodologically or empirically, and backpropagating through Algorithm 1 may be non-trivial.

**Essential References Not Discussed:**

N/A

**Experimental Designs Or Analyses:**

Please see "Methods And Evaluation Criteria" above.

**Methods And Evaluation Criteria:**

1. The authors have only empirically evaluated the estimation setting (Section 2) in their experiments, but not the learning setting (Section 4). The authors should evaluate their method on the learning setting as well, particularly when the hypothesis class is infinite and gradient-based methods are required (as the authors discuss at the end of Section 4).

2. The authors have only tested on synthetic data with fairly simple ground truth signals. It would be interesting to test the learning setting on real time-series such as those in the Wild-Time dataset.

**Other Comments Or Suggestions:**

Typos: "syetm" on L147,  "gven" on L254

**Other Strengths And Weaknesses:**

N/A

**Questions For Authors:**

1. Could the authors give some intuition on how they are able to achieve Lemma 1 and Theorem 2 without any assumptions on $\theta_i$, e.g. bounds on $|\theta_i|$ or $|\theta_{i+1} - \theta_i|$?

**Relation To Broader Scientific Literature:**

The authors study a variant of the problem proposed in Hanneke and Yang (2019) for learning in the nonstationary sequential setting. Their primary baseline is the work by Mazzetto and Upfal (2023), which proposes a sliding window algorithm. The authors propose an algorithm based on wavelet transforms which has a long history in time series analysis.

**Theoretical Claims:**

I did not check the proofs of the theorems.

---

> ### Author Rebuttal · Authors · 2025-03-31
>
> We thank the reviewer for their comments. Please see the responses below.
>
> **Why DB8 can outperform Haar wavelets in the synthetic experiments**
>
> The main reason why higher order wavelets can outperform Haar is because, $k+1$-th order wavelets provide an ideal basis for sparsely compressing the information in a gorundtruth signal that has a piecewise polynomial trend with degree upto $k$. A sparse set of wavelet coefficients of the groundtruth signal leads to a low value for the bound in Lemma 1. We attempt to capture this intuition empirically as demonstrated in Fig.4 in Appendix and pointed the readers to that figure in the main text itself (L165, col 1).
>
> **Theoretical justification for using higher order wavelets**
>
> Wavelet basis of order $k+1$ is known to be an ideal basis for sparsely representing piecewise polynomials of degree (upto) $k$, or more generally groundtruths with low $k$-th order total variation. $k$-th order total variation measures the the variation incurred in the $k$-th order (discrete) derivatives of a groundtruth. It is known in the literature that low $k$-th order TV for a groundtruth is equivalent to sparse (measured in the sense of L1 norm) wavelet coefficients in a $k+1$-th order wavelet basis. For a precise quantification see Theorem 1 in [4].
>
> **Extension to the multivariate setting**
>
> We acknowledge that we study the estimation in a univariate setting. Extension of our point-wise estimation proposal to multivariate time series that takes into account inter-task correlations is an interesting direction to explore as a future work.
>
> **Experiments on real dataset**
>
> Please see the response to Reviewer c9dT under the same comment title.
>
> **Intuition behind achieving Lemma 1 and Theorem 2 without assuming prior bounds on $|\theta_i|$ or $|\theta_{i+1} - \theta_i|$**
>
> We first address informally why no prior bound on $|\theta_i|$ is required. The wavelet soft-thresholding estimator applies a soft-threshold at the level of $\tau := \sigma \sqrt{\log n}$. Suppose $\alpha_i$ is a wavelet coefficient of the groundtruth data. Consider the case when $|\alpha_i| > \tau$. Then the shrinkage caused by soft-thresholding only introduces a bias of at-most $\tau$. Similarly when  $|\alpha_i| \le \tau$, then the bias is capped at $|\alpha_i|$ itself. This intuition can be extended to noisy wavelet coefficients upto constants using concentration arguments.
>
> Eventhough we do not assume any prior knowledge on $|\theta_{i+1} - \theta_i|$, note that the bound in Theorem 2 naturally gets worse when there is lot of intra-sequence total variation. We remark that in the bound $\bar theta_{1:t} = (\theta_1 + \ldots + \theta_t)/t$. Hence higher intra-sequence total variation leads to larger values for the bound. The highlight here is that the bound (which is minimax optimal as per the results of Mazzetto and Upfal 2023) in Theorem 2 is obtained with no such prior knowledge on the intra-sequence variation. The reason for attaining such an adaptive bound is primarily algebraic as highlighted in the proof of Theorem 2.
>
>
> **References**
>
> [4] Minimax Estimation via Wavelet Shrinkage, David L. Donoho and Iain M. Johnstone, Annals of Statistics, 1998

---

### Official Review · Reviewer_ZcGb · 2025-03-20

**Overall Recommendation:** 4

**Summary:**

Given noisy observations of independent but non-identical random variables, the authors consider the problem of estimating the most recent ground truth.

A key insight of the paper is that although the ground truth sequence may be non-stationary in the time domain, its wavelet transform reveals a sparse structure. Leveraging this sparsity, the authors propose a wavelet soft-thresholding (denoising) algorithm that automatically adapts to the level of temporal shift. In addition to providing pointwise error bounds for this estimator, the paper extends the ideas in two important directions:

- It analyzes the effect of temporal distribution shift on the performance of machine learning models via upper and lower bounds on the loss function.
- It applies the estimation method to binary classification under distribution shift—developing a computationally efficient ERM-based algorithm that uses a single call to an ERM oracle yet achieves near-optimal excess risk bounds.

Finally, the paper draws connections between its estimation error guarantees and the classical problem of total-variation (TV) denoising, which shows that any algorithm achieving similar pointwise error guarantees is minimax optimal for TV-denoising.

**Claims And Evidence:**

Yes I found the claims to be well supported.

**Essential References Not Discussed:**

N/A

**Experimental Designs Or Analyses:**

The algorithm performs well on the simulations. Although the baselines are better on a certain class of ground-truth signals, the proposed approach looks reasonably robust (I suspect the performance drop is due to the ground truth signal not being smooth enough).

**Methods And Evaluation Criteria:**

I found the methodology to be well formulated. Specifically the authors use the following set of ideas

- Wavelet Representation:
  Although the ground truth $\theta_1, \dots, \theta_n$ may be nonstationary in the time domain, its wavelet transform reveals a sparse structure; only a few key wavelet coefficients carry most of the information.

- Wavelet Denoising Algorithm:
  The proposed algorithm (Algorithm 1) proceeds as follows:
  1. Wavelet Transform: Compute the empirical wavelet coefficients:
     $\tilde{\beta} = W y$
     where $y = [y_n, \dots, y_1]^T$ and $W$ is the wavelet transform matrix.

  2. Soft-thresholding: Apply soft-thresholding to the coefficients:
     $\hat{\beta} = T_\lambda(\tilde{\beta}), \quad \text{with} \quad T_\lambda(x) = \operatorname{sign}(x)\max\{|x| - \lambda, 0\}.$

  3. Reconstruction: Obtain the denoised signal by the inverse wavelet transform:
     $\hat{\theta} = W^T \hat{\beta}.$
     The final estimate $\hat{\theta}_1$ is the last coordinate of $\hat{\theta}$.

**Other Comments Or Suggestions:**

- The rate in Theorem 11 has $n^{\frac{1}{3}}$ for the square loss and $n^{\frac{2}{3}}$ for the absolute error. Have these been interchanged?

- The paragraph below corollary 8 needs to be rewritten. There are many missing details and typos, such as "When $\beta$, the lower
bound reduces to the conclusion that the loss is larger than 0"

**Other Strengths And Weaknesses:**

N/A

**Questions For Authors:**

N/A

**Relation To Broader Scientific Literature:**

- I think this paper makes a significant contribution to the theory of learning under distribution shifts by leveraging classical techniques from Donoho.

- The error guarantees are pointwise, which is stronger than typical cumulative error metrics in online learning.

- The algorithm is also computationally feasible as it can leverage fast Wavelet transforms.

**Theoretical Claims:**

I did not check the proofs in detail, but the results are believable given prior classic work.

 - Pointwise Error Bound:
  For the Haar wavelet system, the paper shows that with high probability,
  $  |\hat{\theta}_1 - \theta_1| \leq \kappa \cdot U(r^*),$
  where $U(r^*)$ reflects the local bias-variance trade-off of the data, and $\kappa$ is a constant (depending on logarithmic factors).

- Adaptivity via Sparsity:
  The error bound is directly linked to the sparsity of the wavelet coefficients. Using higher order wavelet systems (e.g., Daubechies DB8) can capture more complex trends, potentially leading to even sharper error bounds.

- Excess Risk for Classification:
  The approach is extended to binary classification under temporal distribution shift. By employing wavelet-based loss estimates, the authors design an empirical risk minimization (ERM) procedure that achieves near-optimal excess risk bounds with only one ERM call.

- Optimality for Total Variation Denoising:
  The paper proves that any algorithm satisfying a pointwise error bound like that of their wavelet estimator is minimax optimal for the TV-denoising problem (up to logarithmic factors).

---

> ### Author Rebuttal · Authors · 2025-03-31
>
> We thank the reviewer for correctly recognizing and appreciating our contributions.
>
> **Rates in Theorem 11**
>
> The displayed rates are indeed the optimal rates for squared and absolute losses.
>
> **Paragraph below corollary 8**
>
> We will fix the typos and rewrite it for better clarity.

---

### Official Review · Reviewer_zp7o · 2025-03-24

**Overall Recommendation:** 3

**Summary:**

In this paper, the authors  study the problem of estimation and learning under temporal distribution shift. In simple words (of the author),  they consider an estimation task where  n independently drawn observations are given by y_n, \hdots, y_1 and E[y_i] = \theta_i. The goal is to construct an estimator for theta_i. The authors do not assume that the observations are identitally distributed.

**Claims And Evidence:**

The authors claim that wavelet-denoising-based algorithm using Haar tranform achieves the optimal point-wise estimation error guarantees, which has been proved for subGaussian noise, which is a favourable case. It has not been demonstrated for other distributions.

The authors demonstrate the effect of temporally shifted distribution in ML settings. However, this is done under assumptions with  infinite number of samples from the training data for perfect model training, and perfect ML training is considered which largely dilutes the problem in the ML setting.

**Essential References Not Discussed:**

NA

**Experimental Designs Or Analyses:**

NA.

**Methods And Evaluation Criteria:**

Experimental results are most lacking in this work. Only synthetic datasets have been used. Demonstration on real-world data sets that have temporally shifted data is useful.  Baselines are not provided and all results are based on ablations.

**Other Comments Or Suggestions:**

Some additional comments are as follows:
1. Here, the data is not modeled as a random process. Since it is a closely related technique, it is appropriate to introduce literature in this direction?
2. In Lemma 1, subGaussian assumption helps achieve faster convergence usually. What happens under other probabilistic assumptions on \epsilon?
3. Lemma 1 holds for the ground truth that has a linear representation using wavelet coefficients. What happens to the upper bound when such a linear representation does not exist. More specifically, it is essential that the authors list the set of assumptions prior to analysis. 4. Several constants in Lemma 1 are not defined. Further, H in algorithm table is not defined.
5. In Theorem 2, what is \bar theta?
6. Lemma 4 and proof is not clear. The proof is adapted from (Achille and Soatto, 2018)  and several notations are not defined.
7. This work attempts to connect the statistical formulation to that of machine learning. However, the main essence of such an extension arises in how the dataset challenges are handled. However assumptions with  infinite number of samples from the train-ing data for perfect model training and perfect ML training are considered which largely dilutes the problem in the ML setting.

**Other Strengths And Weaknesses:**

The problem addressed is interesting and important connections between the statistical literature and machine learning formulation has been made. However, several simplifying assumptions hider the true potential of the work. The major weakness of the work are the experimental results as detailed above.

**Questions For Authors:**

Same as above.

**Relation To Broader Scientific Literature:**

The problem presented by the authors is very interesting as it takes an alternate route to analysing temporally shifted data. One way to model such a data would be to assume stationarity and fit a random process. However, this work does not assume stationarity or identical distributions. The modelling is achieved using wavelet transforms which is a more general formulation. The stationarity of the sequence in turn affects the sparsity of the wavelet coefficients.

**Theoretical Claims:**

Some of the proofs are checked and following are the issues:

1.. Lemma 1 holds for the ground truth that has a linear representation using wavelet coefficients. What happens to the upper bound when such a linear representation does not exist. More specifically, it is essential that the authors list the set of assumptions prior to analysis. Several constants in Lemma 1 are not defined. Further, H in algorithm table is not defined.
2. In Theorem 2, what is \bar theta?
3. Lemma 4 and proof is not clear. The proof is adapted from (Achille and Soatto, 2018)  and several notations are not defined.

---

> ### Author Rebuttal · Authors · 2025-03-31
>
> We thank the reviewer for their comments. Please see the responses inline.
>
> **Experiments on real dataset**
>
> Please see the response to Reviewer c9dT under the same comment title.
>
> **Experiments are based on ablation**
>
> Ablation on noise level was conducted primarily to test the validity of the algorithm for various signal-to-noise levels.
>
> **Existence of Linear Representation of Ground Truth**
>
> Wavelet basis are universal basis for $\mathbb{R}^n$. This means that they are of full rank and any ground-truth sequence in $\mathbb{R}^n$ can be written as a linear combination of the basis vectors. So the existence of such linear combination is not an assumption, but a structural property that always holds true.
>
> **Extending to other noise distributions**
>
> In this work we primarily focused on subgaussian noise, as the main motivation for our methods is to get high quality estimates of the groundtruth without any prior knowledge on how it evolves. This is common in many studies from non-paramteric regression (see for eg [6] for a comprehensive exposition).  That said, extending to other noise distributions that are heavy tailed is a possible future direction of research.
>
> **Missing Constants in Lemma 1**
>
> We believe that we have defined all constants pertaining to Lemma 1 in its statement.
>
> **$H$ in Algorithm 1**
>
> Thanks for pointing this typo, instead of $H$, it should be the Wavelet Transform Matrix $W$ that is given as input to the algorithm.
>
> **Meaning of $\bar \theta_{1:t}$**
>
> It is the average of last $t$ ground truth values. $\bar \theta_{1:t} := (\theta_1+\ldots+\theta_t)/t$. We will update the manuscript to reflect this.
>
> **Assumptions for estimation**
>
> The only assumption we make is the fact that the noise in the observations are iid sub-gaussian as stated in Lemma 1. No further assumptions are made.
>
> **Connection to random process**
>
> Random processes such as Gaussian processes impose prior structural assumptions (which are often hard to verify) on how the ground-truth evolves over time. However, no such structural assumptions are placed while deriving our bounds in Lemma 1 and Theorem 2. Instead, the bound naturally adapts to the degree of smoothness in the ground-truth. We will mention this in the context of random process.
>
> However, viewing our problem setup through the lens of GPs holds the potential to unlock new properties that are simultaneously connected to estimation quality and uncertainty quantification. This is indeed a good direction to explore in the future.
>
> **The proof is adapted from (Achille and Soatto, 2018)**
>
> In the paper, we have acknowledged in Appendix C that we adapt the proof from (Achille and Soatto, 2018). Note that (Achille and Soatto, 2018) did not make a distinction between the training distribution $D_{Tr}$ and the predictive distribution $p_{\theta}( y | \mathbf{x} )$.
> As the result, it is important to include our full proof so that avid readers are able to understand and recreate all of the mathematic derivations. The proof will also serve as a reference for researchers who want to improve our performance bounds in Theorem 7.
>
> **Infinite number of training data and perfect ML model training in Section 3**
>
> Our goal with Section 3 was just to act as a compelling motivation for Section 4. In particular, to formally show how distribution shift between training and test can degrade model performance. Though we deal with population level losses, extension to finite samples can be realized with standard concentration inequality arguments. Our focus was just to show formally that if the test distribution is a mixture of training and some other contaminant distribution, then the model performance degrades as the contamination fraction increases.
>
> This sets the stage that performance degradation is inevitable even in an ideal case for perfect model training and it is important to develop algorithms whose performance degrades gracefully under distribution shift in practical use-cases. Note that to develop our algorithms in Sections 4, we already considered realistic scenarios with limited training data and imperfect machine learning training processes.
>
> **References**
>
> [6] Adaptive Piecewise Polynomial Estimation via Trend Filtering, Ryan Tibshirani, Annals of Statistics 2014

---

### Decision · Program_Chairs · 2025-05-01

**Decision:**

Accept (poster)

**Comment:**

**Summary.**

The topic of this work is estimation under temporal distribution shift.
Specifically, after observing a sequence of n independent but not identically distributed $Y_t$, each with mean $\theta_t$, the goal is to estimate the “last” parameter $\theta_1$ and to bound the point-wise performance guarantee.

The authors propose to use an algorithm based on wavelets and show that it achieves comparable performance with the prior work of  Mazzetto and Upfal (2023) when specialized to use the Haar wavelet system. Other wavelet systems may even yield sharper bounds, and which adapt to the sparsity of the wavelet coefficients.

They propose an analysis of the detrimental effects of distribution shift for machine learning models.

They obtain additional results in the context of binary classification and an optimality condition for the problem of TV denoising.

Finally, the authors demonstrate the superiority of they algorithm based on wavelets through experiments (synthetic data).

**Strengths.**

* The contribution was deemed significant (ZcGb); the problem considered is foundational (c9dT).
* The method is computationally tractable (ZcGb); in particular, it is more computationally efficient than Mazzetto and Upfal (2023).
* The obtained pointwise guarantee is strong (ZcGb).

**Weaknesses.**

* The analysis is limited to sub-Gaussian noise (zp7o).
* The analysis is restricted to the univariate setting (4jCD).
* No experiments for the learning setting (4jCD).
* No principled method for choosing the optimal wavelet transform (8JTP).

**Discussion and reviewer consensus.**

ZcGb did not acknowledge the rebuttal and 8JTP did not share their updated thoughts after the author response. I think the authors addressed the concerns of 8JTP satisfactorily; they also acknowledged that optimal wavelet selection remains a limitation of their work.

**Additional remarks.**

* Originally, there was a concern about the absence of experiments with real data (c9dT, 8JTP, zp7o), but in their response, the authors went the extra mile and ran additional experiments with data from the Dubai Land Department, which was well-received be the referee (c9dT, zp7o).
* In addition, a certain amount of typos were reported by the reviewers; should the paper be accepted, I recommend the authors to carefully address them.

**Overall evaluation.**

I recommend acceptance if there is room in the program.